# The transcriptomic response of cells to a drug combination is more than the sum of the responses to the monotherapies

Jennifer EL Diaz[1,2,3†], Mehmet Eren Ahsen[1,3,4†‡], Thomas Schaffter[1,3§], Xintong Chen[1], Ronald B Realubit[5,6], Charles Karan[5,6], Andrea Califano[5,7,8,9], Bojan Losic[1,10,11,12], Gustavo Stolovitzky[1,3,5,7]*

[1]Department of Genetics and Genomics Sciences, Icahn School of Medicine at Mount Sinai, New York, United States; [2]Department of Cell, Developmental, and Regenerative Biology, Icahn School of Medicine at Mount Sinai, New York, United States; [3]IBM Computational Biology Center, IBM Research, Yorktown Heights, United States; [4]Department of Business Administration, University of Illinois at Urbana-Champaign, Champaign, United States; [5]Department of Systems Biology, Columbia University, New York, United States; [6]Sulzberger Columbia Genome Center, High Throughput Screening Facility, Columbia University Medical Center, New York, United States; [7]Department of Biomedical Informatics, Columbia University, New York, United States; [8]Department of Biochemistry and Molecular Biophysics, Columbia University, New York, United States; [9]Department of Medicine, Columbia University, New York, United States; [10]Tisch Cancer Institute, Cancer Immunology, Icahn School of Medicine at Mount Sinai, New York, United States; [11]Diabetes, Obesity and Metabolism Institute, Icahn School of Medicine at Mount Sinai, New York, United States; [12]Icahn Institute for Data Science and Genomic Technology, Icahn School of Medicine at Mount Sinai, New York, United States

*For correspondence:
gustavo@us.ibm.com

†These authors contributed equally to this work

Present address: ‡Department of Business Administration, University of Illinois at Urbana-Champaign, Champaign, United States; §Computational Oncology Group, Sage Bionetworks, Seattle, United States

**Abstract** Our ability to discover effective drug combinations is limited, in part by insufficient understanding of how the transcriptional response of two monotherapies results in that of their combination. We analyzed matched time course RNAseq profiling of cells treated with single drugs and their combinations and found that the transcriptional signature of the synergistic combination was unique relative to that of either constituent monotherapy. The sequential activation of transcription factors in time in the gene regulatory network was implicated. The nature of this transcriptional cascade suggests that drug synergy may ensue when the transcriptional responses elicited by two unrelated individual drugs are correlated. We used these results as the basis of a simple prediction algorithm attaining an AUROC of 0.77 in the prediction of synergistic drug combinations in an independent dataset.

## Introduction

Combination therapy has become increasingly relevant in cancer treatment (*Al-Lazikani et al., 2012*; *Ellegaard et al., 2016*). The complexity of patient-to-patient heterogeneity (*Palmer and Sorger, 2017*), intratumoral heterogeneity (*Gerlinger et al., 2012*), and intracellular pathway dysregulation (*Hyman et al., 2017*) provides opportunities for combining drugs to induce responses that cannot be achieved with monotherapy. Effective combinations may target multiple pathways (*Larkin et al., 2015*) or the same pathway (*Baselga et al., 2012*). They may also reduce the dose of individual

drugs, thereby reducing toxicity, or target molecular mechanisms of resistance, thereby prolonging the effective duration of treatment (*Al-Lazikani et al., 2012*; *Greco et al., 1996*; *LoRusso et al., 2012*; *Lehár et al., 2009*).

Drug combinations are said to be synergistic if their activity exceeds their expected additive or independent response (*Palmer and Sorger, 2017*; *Greco et al., 1995*). Synergistic behavior is difficult to predict, so rational combinations may not validate experimentally (*Wolff et al., 2013*). Hypothesis-driven studies of the mechanisms of synergy and antagonism have focused on a limited set of candidate targets (*Bollenbach et al., 2009*; *Jiang et al., 2018*). Alternatively, unbiased high-throughput screening assays (*Mott et al., 2015*; *Mathews Griner et al., 2014*; *Borisy et al., 2003*) can identify synergistic compounds in a systematic way by assessing cell viability reduction by individual drugs and their combinations. Unfortunately, screening all possible drug-pairs in a panel of N drugs with $N_C$ cell lines at $N_D$ doses requires a large number ($\frac{1}{2}$ N (N-1) $N_D^2$ $N_C$) of experiments, resulting in high costs that limit the practical reach of this approach. Computational methods to predict synergistic combination candidates are needed to improve the experimental cost-benefit ratio (*Fitzgerald et al., 2006*; *Bansal et al., 2014*).

To address this need, the DREAM (Dialogue on Reverse Engineering Assessment and Methods) Challenges consortium (*Saez-Rodriguez et al., 2016*) conducted two community-wide competitions (the NCI-DREAM Drug Synergy Prediction Challenge, *Bansal et al., 2014*; and the Astra Zenca-Sanger Drug Combination Prediction DREAM Challenge, *AstraZeneca-Sanger Drug Combination DREAM Consortium et al., 2019*) that fostered the development and benchmarking of algorithms for drug synergy prediction. In the the NCI-DREAM Drug Synergy Prediction Challenge, the organizers provided post-treatment transcriptomics data for each of 14 drugs administered to a lymphoma cell line, and asked participants to predict which of the 91 pairwise combinations would be synergistic (*Bansal et al., 2014*; *Sun et al., 2015*). One of the key outcomes of the Challenge and other studies was that synergistic drug combinations could be partially predicted from the transcriptomics of the monotherapies (*Sun et al., 2015*; *Niepel et al., 2017*). The two best-performing teams based their algorithms on the assumption that a concordance of gene expression signatures in drugs with different mechanisms of action often yield synergistic interactions (*Yang et al., 2015*; *Goswami et al., 2015*). This assumption, while plausible, has little experimental support beyond winning the Challenge. Further, the mechanism behind this phenomenon cannot be ascertained without transcriptomics data from the combination therapy, which was not provided in the Challenge due to cost. We therefore pose a fundamental question that can only be answered with matched monotherapy-combination transcriptomics data: *How do two different transcriptomics profiles in cells treated with two different drugs combine to give a new transcriptomics profile when the drugs are applied together?* If the combinatorial pattern of two gene expression profiles are different in synergistic versus additive drug combinations, then learning to recognize these patterns may enable us to predict synergistic combinations from the gene expression of monotherapies.

In this paper, we explore the relationship between the transcriptional landscape of drug combinations in relation to the profiles of the individual drugs. We performed a systematic, genome-wide analysis of matched time courses of gene expression following perturbation with individual compounds and with their combinations. Deliberately sacrificing breadth for depth, we studied the transcriptional temporal response of an empirically chosen synergistic drug combination, tamoxifen and mefloquine, in breast cancer and prostate cancer cells and compared it to that of additive combinations of withaferin with either tamoxifen or mefloquine. Rather than elucidating specific mechanisms of action for drugs and their combinations, we attempt to examine the transcriptome for molecular indicators of synergy. Our analysis shows that molecular synergy (measured by the number of genes whose expression changes significantly only in the combination), correlates with the Excess Over Bliss independence, a measure of the observed effect of a combination that is greater than the expected effect based on the Bliss model of additivity (*Greco et al., 1995*) and increases with time. We used network-based analyses to trace the transcriptional cascade as it unfolds in time in the synergistic combination. We found that transcription factors simultaneously activated by both drugs dominate the cascade. We propose that a pair of drugs with correlated expression signatures is likely to trigger a synergistic effect, even when they target different pathways. We contrast this effect with the correlated, but additive, effect of increasing dose of one drug. Correlation of monotherapies predicted synergistic drug interaction with relatively high accuracy (AUROC = 0.77) using

the independent DREAM dataset. This study represents a matched monotherapy-combination transcriptomic analysis of synergy, advancing both our understanding of synergy and ability to predict it.

The paper is organized as follows: First, we describe why we chose the monotherapies and combinations used in this paper and identify patterns of gene expression across synergistic and additive combinations. We analyze how those molecular patterns relate to phenotypic synergy and explore synergistic effects on biological processes over time with drug treatment. We then describe our exploration of synergistic effects on gene splicing as an alternative mediator of drug combination effect. Returning to gene expression, we study the mechanism of synergistic gene expression changes by identifying differentially active transcription factors through a transcriptional network and tracing the impact of these transcription factors in a temporal activation cascade. Then, we use an independent microarray dataset to verify the hypothesis that correlation between gene expression profiles of monotherapies can be used as an indicator of synergy. Finally, we discuss the structure of the synergistic transcriptional cascade and a plausible conceptual framework for the molecular underpinnings of synergy.

## Results

### Finding reproducible synergistic and additive combinations

To identify drug-pairs for the detailed time course RNA-seq analysis, we leveraged a pre-existing LINCS drug combination dataset collected at Columbia University (Califano's lab) in the MCF7 cell line, a breast cancer line that is ERα positive (*Lee et al., 2015*), and the LNCaP cell line, a prostate cancer cell line that is ERβ positive (*Takahashi et al., 2007*). This dataset had information on all combinations of 99 drugs against 10 different drugs, each combination assessed in a matrix of 4 by 4 doses at 48 hr after drug application. Among these 990 drug combinations, we found 39 synergistic combinations with a maximum Excess Over Bliss (EOB) independence over the $4 \times 4$ matrix of more than 30%. From these 39, to select synergistic combinations in which both monotherapies played an important role in eliciting synergy, we chose 13 combinations whose constituent monotherapies showed a variety of combinatorial behaviors across the combinations (antagonism, additivity, and synergy in different combinations) for further testing. We re-assessed the synergy of these 13 pairs using a $6 \times 6$ dose response matrix and found that 9 of the 13 combinations remained synergistic, while four exhibited additive or antagonistic responses, which we removed from further study. Tamoxifen appeared in the greatest number (four) of these nine combinations. Given this data and the known clinical utility of Tamoxifen in ER+ breast cancer (*Davies et al., 2011*; *Early Breast Cancer Trialists' Collaborative Group (EBCTCG), 2015*), we focused on these four combinations. Of those 4, we sub-selected the two drug pairs with the highest EOB values, Tamoxifen (T) + Mefloquine (M) and T + Withaferin A (W) (EOB = 49 and 43, respectively). We then further measured viability (using the high-throughput Cell Titer Glo) and EOB in triplicate experiments using a $10 \times 10$ dose matrix at 12 hr, 24 hr and 48 hr (*Supplementary files 1–2*) after drug treatment, obtaining results that were consistent with the previously found synergy. Note that these two combinations were tested in at least three independent sets of experiments at this point (99 × 10 screen, 13 combinations, and 2 combinations).

We noted differences in viability, consistent with recent concerns regarding the lack of reproducibility in cell line viability experiments in response to drugs (*Haibe-Kains et al., 2013*; *Ben-David et al., 2018*), and yet TM remained consistently synergistic despite changes in the viability of its constituent monotherapies. In addition, we noted hormetic dose curves in response to the monotherapies, especially for W (*Mattson and Calabrese, 2010*). These hormetic responses are evidenced by a non-monotonic dose response, with more than 100% viability with respect to control for small doses (about 3 uM for W; *Supplementary file 2*) or at short times after drug application (T and M at 3 hr, *Figure 1C–E*). Many factors could contribute to hormesis. For example, it has been observed that efficient use of energetic processes in complex stress responses require biological resources to be deployed by the cell in a timely fashion (temporal hormesis) and at relatively low-doses (dose hormesis) to elicit a protective response (*Li et al., 2019a*; *Mattson, 2008*). The elucidation of these hormetic responses in the context of synergistic interactions could be a fruitful line of research, but goes beyond the scope of this paper.

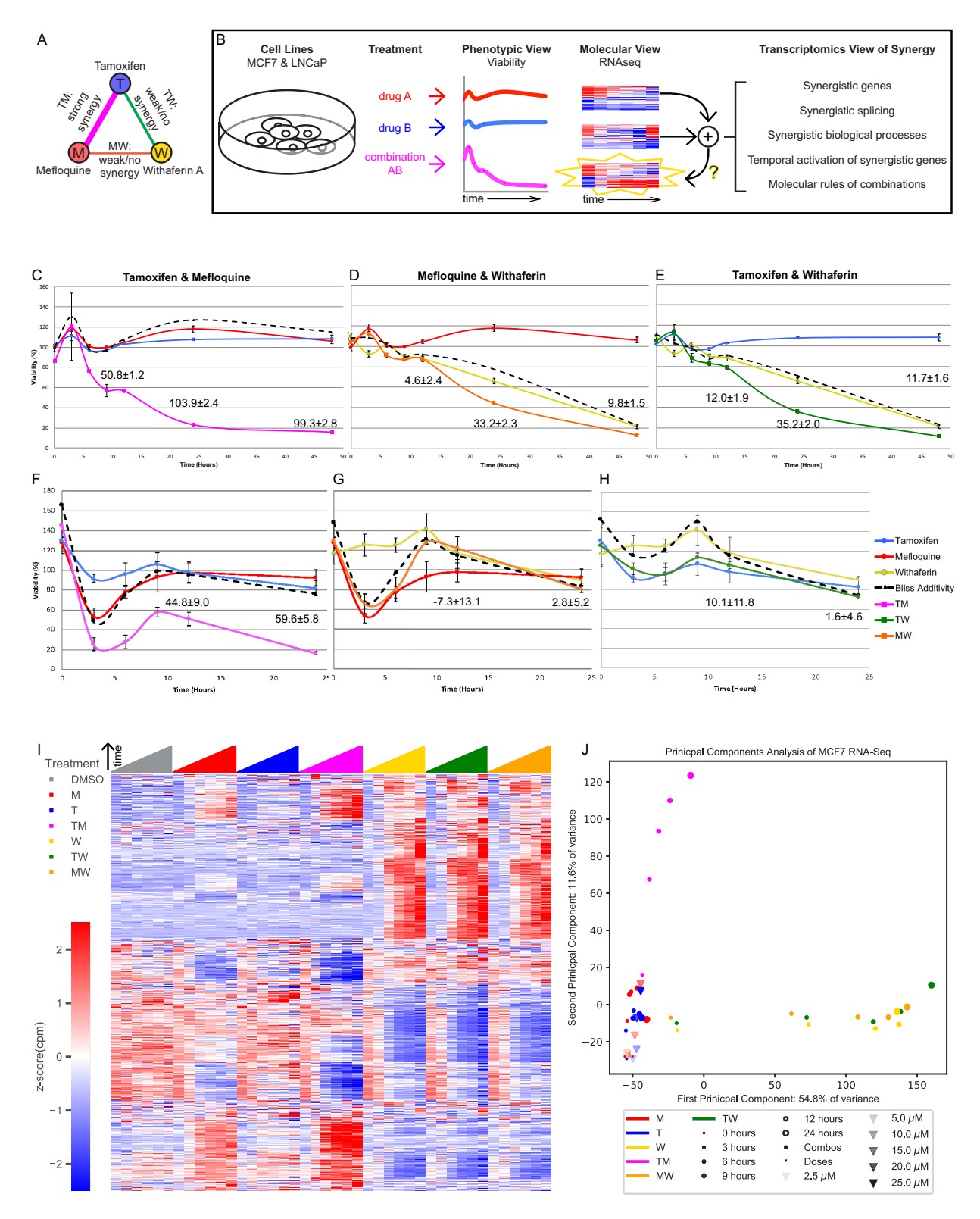

**Figure 1.** The transcriptomics of drug combinations mirror their phenotypic characteristics. (**A**) Monotherapies and drug combinations used in the study. (**B**) Workflow of molecular analysis of synergy. Starburst highlights the novel component of RNAseq analysis. Question mark denotes the focus of the study. (**C–H**) Fold change over control of cell count for MCF7 cells (**C–E**) and LNCaP cells (**F–H**) treated with Tamoxifen and Mefloquine (**C,F**), Mefloquine and Withaferin (**D,G**), and Tamoxifen and Withaferin (**E,H**). Dashed line indicates predicted viability of the combination based on the Bliss

*Figure 1 continued on next page*

*Figure 1 continued*

model. Excess Over Bliss (EOB) ± Error$_{EOB}$ is given for the 12, 24, and 48 hr time points (see Materials and methods). (I) Average gene expression for each treatment and time point in the MCF7 combination experiments (covering 108 treatment and 18 DMSO samples). (J) Principal component analysis of log fold change in gene expression vs DMSO for the average over replicates at each treatment and time point in the MCF7 combination and dose experiments. (See also *Supplementary files 3–9*, and *Source data 1*).

The online version of this article includes the following figure supplement(s) for figure 1:

**Figure supplement 1.** The response of MCF7 to TM is more synergistic than to TW.
**Figure supplement 2.** Transcriptomic profiles of MFC7 and LNCaP cells with combinations.

Finally, we selected doses of these three drugs that synergized in these two combinations at both time points (20 μM T, 10 μM M, 5 μM W) for subsequent study (*Supplementary files 3–4*). T and M are also synergistic at 12 and 24 hr as measured by combination index, which quantifies synergy factoring out the dose effects, but T and W are synergistic only at 24 hr (*Figure 1—figure supplement 1A*). For completeness, we included the combination MW in subsequent studies.

We focused the rest of the study on these three drugs and their combinations (*Figure 1A*), in an effort to understand how synergy operates at a transcriptomic level (*Figure 1B*). We studied MCF7 and LNCaP cells under DMSO (vehicle control), T, M and W, and their combinations TM, TW and MW over a 48 hr time course (0, 3, 6, 9, 12, 24 and 48 hr) using nuclei counts as a direct readout of cell viability in relation to DMSO (*Figure 1C–H*). At these doses, TM (*Figure 1C,F*) synergistically reduced viability as early as 6 hr, with little effect from T and M individually in MCF7 (*Figure 1C*) and moderate effect in LNCaP (*Figure 1F*). The synergistic effect of TW and MW was very small compared to TM in MCF7 (*Figure 1D,E*) and negligible in LNCaP (*Figure 1G,H*). In relation to TM, we therefore consider the effects of TW and MW to be additive and dominated by W (*Supplementary files 2*, *5*).

Finally, we studied the effect of drug dose on viability, as a combination treatment exposes the cells to more drug than a monotherapy, and this could mimic the effect of increasing drug dose. Additionally, M has been shown to inhibit the function of MDR1, a multi-drug efflux pump (*Riffkin et al., 1996*) and its effect could therefore be simply to increase the intracellular tamoxifen concentration. We analyzed dose curves of T and M as monotherapies (*Figure 1—figure supplement 1B*). We measured the effect of T alone at 5, 10, 20, 25, and 30 μM, and M alone at 2.5, 5, 10, and 15 μM at 24 hr in MCF7 cells. Viability in 25 and 30 μM T (37.1% and 13.7%) was similar to TM (23%), while viability of cells treated with M at 15 μM was 63.3%. We continued to observe some inter-experiment variability in the efficacy of monotherapies (e.g. T at 20 μM at 24 hr in *Figure 1C* and *Figure 1—figure supplement 1B*, the latter measured about 2 years after the former one). Interpreting 25 μM T as a 'sham' combination of 5 and 20 μM T (*Figure 1—figure supplement 1B*), 30 μM T as a 'sham' combination of 10 and 20 μM T, and 15 μM M as a 'sham' combination of 5 and 10 μM M, we observed EOBs of 31.5, 53.1, and 17.5, respectively (*Supplementary file 6*), far lower than the EOB of about 103.9 in TM (*Figure 1C*). Consistent with the synergistic Combination Index in TM (*Figure 1—figure supplement 1A*), this suggests that the synergy we observed is a phenomenon distinct from dose response. To study the transcriptional mechanisms of drug combinations, we collected RNA from the same cultures from which we measured viability at each treatment for RNA-seq (except 30 μM T, which caused too much cell death for RNA collection).

## Gene expression of drug combinations in relation to monotherapies

For each treatment (in doses and combinations listed above) and time point up to 24 hr, we collected samples in triplicate and performed RNAseq studies (*Supplementary files 7–8*). The RNAseq data was reliable, with replicates showing a very high concordance and technical noise considerably smaller than the changes in expression observed under different conditions (*Figure 1—figure supplement 2A–B*). We first examined the gene expression over all treatments and time points in combination experiments (*Figure 1I*) and variable doses used for the 'sham' combinations (*Figure 1—figure supplement 2C*) in MCF7, and combination experiments in LNCaP (*Figure 1—figure supplement 2D*). The transcriptional profiles for the monotherapies T and M were more similar to DMSO than TM. The transcriptional profiles for W, TW, and MW on the other hand, were similar to each other but different from T, M, TM, and DMSO over time. However, gene expression profiles from

different doses of the same monotherapy were quite similar, with changes evolving gradually with increasing dose (*Figure 1—figure supplement 2C*). This pattern mirrors the phenotypic viability profiles (*Figure 1C–H*, *Figure 1—figure supplement 1B*). *Figure 1J* shows a two-dimensional principal component analysis (PCA) of the transcriptional data from the combination and dose experiments in MCF7. The data for T and M monotherapies, from both the combination experiments and the dose experiments, localize slightly but distinctly above DMSO. However, their TM combination is farther from DMSO than T or M but in the same vertical direction. The PCA representation of W progresses in an almost horizontal direction, with TW and MW co-localizing with W, which dominates the combination. Therefore, the two-dimensional PCA representation of the transcriptomes after treatment suggests orthogonal synergistic and additive directions. The PCA representation of the gene expression from treated LNCaP cells indicates very similar dynamics in this distinct cell line (*Figure 1—figure supplement 2E*).

We next examined differential expression relative to DMSO. The high concordance of replicates allowed for clear detection of differentially expressed genes (DEGs) in different conditions (*Figure 2—figure supplement 1A*). To determine DEGs in MCF7, we selected a false discovery rate (FDR) cutoff at which the only DEGs at time 0 (~30 min post-treatment; see Materials and methods) over all treatments are well-known immediate early genes (*Sas-Chen et al., 2012*; *Tullai et al., 2007*; *Table 1*). We then used this cutoff across all time points and treatments of the fixed dose experiments. To achieve consistency in our treatment of the variable dose experiments which were done separately and with fewer replicates, we chose an FDR cutoff resulting in approximately the same number of DEGs in M 10 µM and T 20 µM (*Figure 2—figure supplement 1B*; see Materials and methods). In LNCaP, we selected anFDR cutoff at which there were no DEGs at time 0, as we noticed no differentially expressed immediate early genes in this case (see Materials and methods). We then quantified and examined the properties of DEGs in monotherapies and their combinations. The number of DEGs in MCF7 cells treated with TM, W, MW, and TW were 1 to 2 orders of magnitude greater than that of treatments with T and M (*Figure 2A–C*). We evaluated the presence of *synergistically expressed genes (SEGs)*, which we define as *genes that are differentially expressed in the combination therapy but not in either of the constituent monotherapies*. Approximately 90% of DEGs in MCF7 cells treated with TM are synergistic, and not differentially expressed in either T or M alone (*Figure 2A*) at any time point. To test for artifacts related to the chosen FDR cutoff, we calculated the percentage of SEGs over different FDR thresholds and observed that the general trend is independent of the specific cutoff (*Figure 2—figure supplement 1C–D*). In contrast, most DEGs in treatments TW and MW were also differentially expressed in W (*Figure 2B–C*). These molecular signatures parallel the effect of these drugs on viability (*Figure 1C–E*), reflecting the overall synergistic character of TM, and a mostly additive dominant effect of W. In LNCaP cells, we observed a similar effect of TM: more than 75% of DEGs are SEGs at any time point (*Figure 2—figure supplement 2A*). However, we observed that when LNCaP cells were treated with TW or MW, more than a quarter of DEGs were SEGs at any time point, and more than half at 12 and 24 hr (*Figure 2—figure supplement 2B–C*); in comparison, in MCF7 cells treated with MW or TW, less than a quarter of DEGs were SEGs at nearly every time point (*Figure 2B–C*). This highlights the

**Table 1.** Selection of adjusted p-value cutoff for differentially expressed genes.

The six most differentially expressed genes with respect to DMSO in each treatment at time 0 are shown in ascending order of their Voom score ($\log_{10}$(FDR)). Immediate Early Genes are marked in red. Differentially expressed genes according to the $1.0 \times 10^{-18}$ cutoff for FDR corrected p-value are marked in bold.

| T_0 | | TM_0 | | M_0 | | TW_0 | | MW_0 | | W_0 | |
|---|---|---|---|---|---|---|---|---|---|---|---|
| FOS | -6 | BCAN | −17 | ATXN2 | -3 | EGR1 | −54 | EGR1 | −53 | EGR1 | −53 |
| MYC | -3 | FOS | −17 | JUN | -2 | JUN | −26 | JUN | −28 | IER2 | −20 |
| TOB1 | -3 | VIM | −17 | ZNF592 | -1 | IER2 | −23 | IER2 | −17 | JUN | −19 |
| KLF4 | -2 | ETS1 | −15 | ZHX2 | -1 | JUNB | −17 | PDCD7 | −17 | C17O | −16 |
| SGK1 | -2 | MSN | −13 | SCAF4 | -1 | PDCD7 | −16 | ZFP36 | −15 | ZFP36 | −16 |
| PRDM1 | -2 | NCAN | −10 | NAT8L | -1 | C17ORF91 | −16 | JUNB | −14 | PDCD7 | −15 |

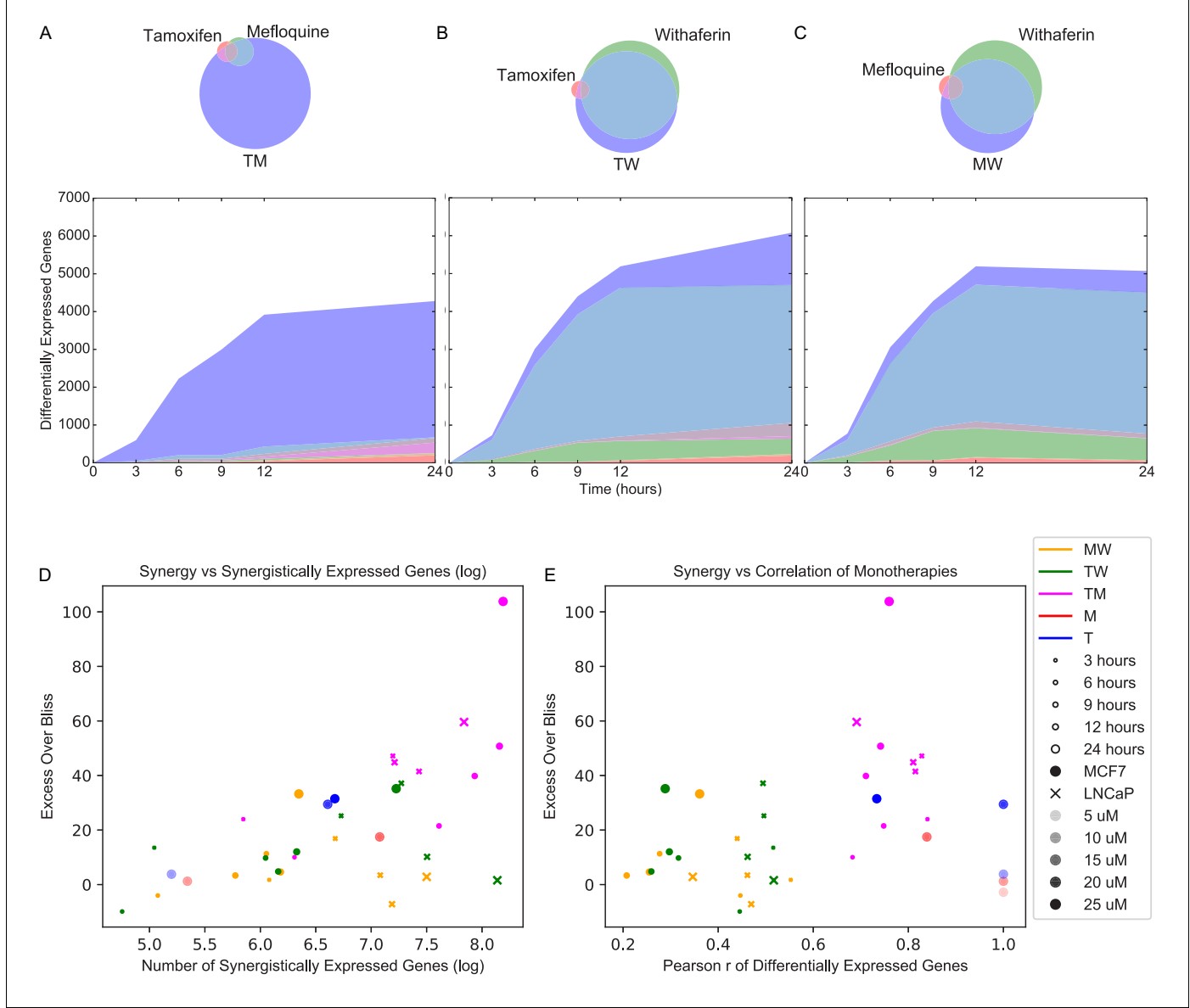

**Figure 2.** Synergistically expressed genes and correlated monotherapies are associated with synergy. (A-C) Number of DEGs over time in MCF7. The Venn diagrams correspond to DEGs at 3 hr in (A) T, M, and TM; (B) T, W, and TW; and (C) M, W, and MW. The area represented in each color is in proportion to the number of genes in the corresponding color of the Venn diagram; blue areas represent SEGs. (D–E) Relationship of Excess Over Bliss score with (D) the number of SEGs, and (E) correlation in gene expression values between each pair of monotherapies. Note that some of the 'pairs' from the dose experiments represent the same dataset correlation with itself (i.e. T 10 μM with T 10 μM for the T 20 μM 'combination') and so have correlation = 1.0 as expected, and are shown for clarity. (See also *Source data 4*, *5*, *6*).

The online version of this article includes the following figure supplement(s) for figure 2:

**Figure supplement 1.** Gene expression characteristics of differential expression and synergy in MCF7.
**Figure supplement 2.** Differential expression in LNCaP Number of DEGs over time in LNCaP.

ability of different cells to respond differently to drugs, and we explore it further in the next section. Interestingly, across both cell lines and including the pairs in our dose experiments, the number of SEGs correlates well with EOB for all treatments and time points, (Pearson r = 0.63, p=0.000044; Spearman $r_s$ = 0.59, p=0.00013; *Figure 2D*). The number of SEGs and the EOB of T 25 μM and M 15 μM are similar to TW and MW and considerably smaller than TM, consistent with the interpretations that behavior of TW and MW represent additivity, and that the molecular and phenotypic synergy of TM transcends the expected behavior of a simple increase in dose.

Finally, we also observed a significant relationship between the EOB of a combination and the correlation of the transcriptional profiles of the constituent monotherapies (*Figure 2E*; see Materials and methods). Conversely, the monotherapy pairs from our dose experiments (i.e. T 5 and 20 µM, M 5 and 10 µM) had high correlation as expected, but low EOB. For this reason, when including the dose experiments, the direct correlation between EOB and the correlation of transcriptional profiles (see Materials and methods) is not significant (Pearson r = 0.31, p=0.068; Spearman $r_s$ = 0.28, p=0.097). When we removed these sham combination pairs from the dose experiments, we observed a significant relationship between EOB and correlation of transcriptional profiles (Pearson r = 0.59, p=0.00064; Spearman $r_s$ = 0.54, p=0.019). This result suggests that correlated transcriptional profiles of two distinct drugs may be important in defining synergy. Further study in other contexts would be necessary to generalize this hypothesis. The possible nature of correlation as a necessary but not sufficient condition for synergy will be discussed further in a later section.

## Critical cancer pathways are synergistically enriched

We checked for enrichment of gene sets associated with specific biological processes. Candidate gene sets were selected from gene-set libraries retrieved from the Enrichr tool (*Chen et al., 2013*; *Kuleshov et al., 2016*), pathways implicated in the hallmarks of cancer (*Hanahan and Weinberg, 2011*), and likely drug targets of T and M (see Materials and methods). *Figure 3* shows the biological processes that are enriched in at least one of the subgroups of DEGs in MCF7 cells under

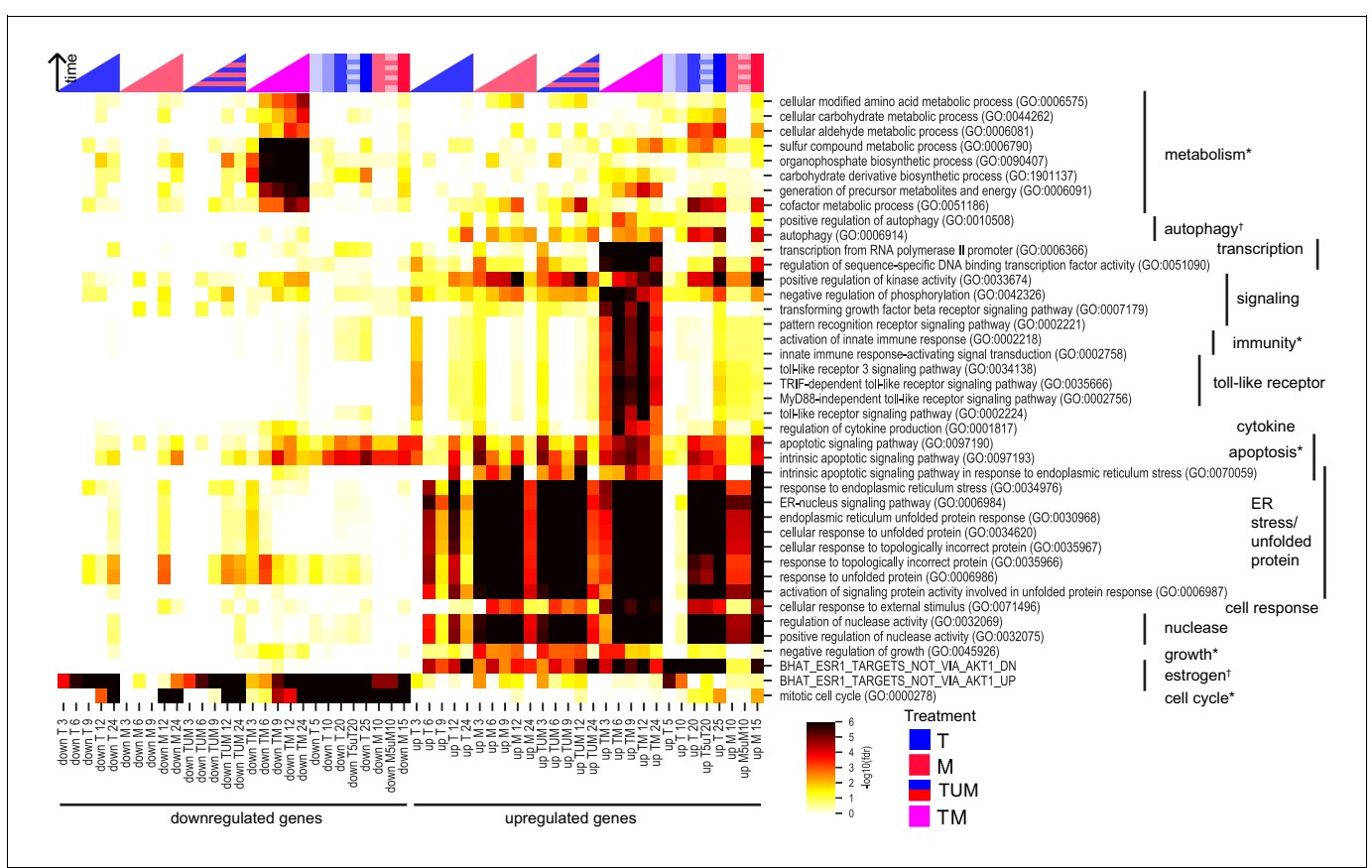

**Figure 3.** Key biological processes are associated with synergy. Enrichment of DEGs in T, M, and TM with cancer-relevant gene sets. Only gene sets enriched in at least one condition (time point or treatment) are shown. 'TUM' indicates the union of DEGs either in T or M. Color intensity reflects degree of enrichment by Fisher's Exact test. Color markers indicate treatment and color marker intensity indicates dose. *=hallmark of cancer, †=drug target (see Materials and methods).

The online version of this article includes the following figure supplement(s) for figure 3:

**Figure supplement 1.** Biological processes in MCF7 and LNCaP.

**Figure supplement 2.** Enrichment in gene sets associated with phospholipidosis (PLD) in MCF7 and LNCaP.

treatment with T, M, TM, as well as in the set TUM, the union of DEGs under T or M, which represents the expected DEGs if T and M acted additively. If T and M act synergistically, we expect that the set of DEGs in TM should be enriched in more functional classes than TUM. Implicated biological processes fell into three classes: (1) endoplasmic reticulum stress, estrogen signaling, and kinase activity were enriched in both TM and monotherapies; (2) apoptosis, toll-like receptor and cytokine signaling, immunity, transcription, metabolic processes, and autophagy were markedly more enriched in the combination TM than in either monotherapy or TUM, an effect not recapitulated by increasing monotherapy dose at 24 hr; and (3) downregulation of the cell cycle was present in both the combination and monotherapies at 12 and 24 hr, but began to occur much earlier in the combination (*Figure 3*). Classes 2 and 3 appear to be synergistically affected in TM but were not synergistic in either TW or MW (*Figure 3—figure supplement 1A*).

We also interrogated the dysregulated genes in the treated LNCaP cells by the same procedure. As more SEGs had appeared in LNCaP cells treated with MW and TW than in MCF7, especially at 12 and 24 hr (*Figure 2D*), we compared the synergistically enriched gene sets in LNCaP cells for TM, TW, and MW (*Figure 3—figure supplement 1B*). A few biological processes, such as autophagy, were synergistically enriched in all three combinations. Gene sets for which we observed differences between the combinations fell into two broad groups: synergistically enriched more in W combinations (W-enriched) or synergistically enriched more in TM (TM-enriched). The W-enriched gene sets included two main classes: (1) cholesterol biosynthesis and metabolism was only synergistically upregulated in MW and (2) rRNA and ncRNA processing was synergistically upregulated only in TW at 24 hr, while tRNA and mitochondrial RNA processing was synergistically downregulated at some time points in both TW and MW. TM-enriched gene sets fell into three classes: (1) temporal differences: endoplasmic reticulum stress was upregulated at earlier time points in TM than in the monotherapies, whereas it was similarly enriched in W, TW, and MW at all time points except 24 hr, and intrinsic apoptosis in response to ER stress was upregulated initially in W, TW, and MW followed by normalization over time, whereas in TM it was synergistically upregulated in an increasing manner over time; (2) certain metabolic processes (generation of precursor metabolites and energy, cofactors, amino acids, and sulfur) were synergistically downregulated only in TM; and (3) genes that are repressed by estrogen receptor were synergistically upregulated only in TM. We hypothesize that the W-enriched classes represent mechanisms by which LNCaP cells counter the effects of the drug combinations and evade cell death, whereas TM-enriched gene sets, particularly class 2, may represent gene sets that function as harbingers of phenotypic synergy, distinguishing synergistic drug combinations from combinations whose effects can be resisted by cells.

Finally, we assessed for enrichment in phospholipidosis (PLD) in both cell lines. Research has shown that drugs that induce lysosomal stress and lipid accumulation (phospholipidosis), including T and M, tend to exhibit similar transcriptional profiles (*Sirci et al., 2017*; *Nioi et al., 2007*; *Nadanaciva et al., 2011*). We quantified enrichment in several types of gene sets with a focus on PLD (*Sirci et al., 2017*; see Materials and methods): cellular components, including in the top 20 gene ontology gene sets associated with PLD; a PLD gene signature (provided by authors of *Sirci et al., 2017*); and a set of the gene targets of two transcription factors (TFE3 and TFEB) shown to be involved in lysosomal stress. We found that some PLD-associated cellular components are synergistically affected in TM, including the lysosome, Golgi, mitochondrion, nucleus, and nucleolus. PLD was highly enriched in all treatments in both cell lines (*Figure 3—figure supplement 2*), indicating generalized toxicity-associated effects of treatment. We studied the role that PLD might play in the high correlation between monotherapies in our experiments. We found that the relationship between correlation and EOB (excluding dose experiments) holds even when PLD genes were removed (r = 0.59, p=0.00068; Spearman $r_s$ = 0.55, p=0.0017). Furthermore, we found that genes in the PLD signature accounted for a small proportion of DEGs in all treatments (data not shown), and as a result the correlation between monotherapies is nearly identical whether we include or exclude the PLD signature genes. (A plot of the correlation between monotherapies including the PLD signature genes vs the correlation between monotherapies excluding the PLD signature genes yielded an almost perfect identity line with: r = 0.9999, p=1e-65; Spearman $r_s$ = 0.9995, p=2e-52.) Finally, enrichment of the PLD signature gene set in TM was only slightly greater than in TUM (*Figure 3—figure supplement 2A* for MCF7, *Figure 3—figure supplement 2C* for LNCaP), indicating at best mild synergy in PLD signature genes. These results show that PLD plays a role in the treatments considered here and that some transcriptional similarity between the monotherapies may be associated

with PLD. However, PLD is one of many cellular processes triggered by the drug treatments, and it accounts only in a small part for the transcriptional correlation and synergistic gene expression we observed.

## Co-expressed genes show a synergistic temporal pattern

We studied temporal patterns of drug response in MCF7. We used k-means clustering (see Materials and methods) to identify co-expressed genes with similar time evolution in T, M, and TM (*Source data 7*). This unsupervised clustering method identified four distinct temporal patterns (*Figure 4A*): (1) upregulated in TM (2253 genes), (2) strongly upregulated in TM (421 genes), (3) downregulated in TM (1709 genes), (4) strongly downregulated in TM (718 genes). In each cluster, the average differential expression observed in the combination TM was significantly stronger than that in T + M, in which the (log) expression in T and M are added. The trajectory over time for most genes is monotonic and saturates at 9 hr. However, we also tested for genes whose trajectories were significantly different in TM than T and M (data not shown). A minority of genes in each cluster exhibited unique temporal profiles in TM, including mixed transient and monotonic behavior, suggesting the existence of temporal synergy (*Figure 4B*).

We then assessed these gene classes for enrichment in biological processes (*Figure 4C*). Consistent with enrichment of these processes at each time point (*Figure 3*), upregulated genes were enriched in endoplasmic reticulum stress (clusters 1–2), and downregulated genes were enriched in cell cycle and metabolic processes (clusters 3–4). In addition, apoptosis and downregulated targets of estrogen were enriched in genes strongly upregulated in TM (cluster 2), highlighting synergistic properties. Metabolic processes and the cell cycle were distinguished by clusters 3 and 4, highlighting the biological significance of the degree of downregulation. Finally, the genes with significantly different trajectories in TM account for a small but distinct subset of these synergistic biological processes (data not shown). Together, these data indicate that monotonic dysregulation dominates gene behavior and triggers important biological processes, which implies that the early transcriptional responses might be sufficient to predict synergy.

## Synergistically spliced and expressed genes are different

We studied splicing by examining the relative exon usage for each gene focusing on MCF7 cells. Combination treatment TM induced unique patterns of relative exon usage, compared to DMSO, T, and M. For example, many exons were less used in TM, consistent with an exon skipping modality of alternative splicing (*Figure 5*). As with differential gene expression, most differentially spliced genes in TM were synergistic, that is, not differentially spliced in either monotherapy (*Figure 6—figure supplement 1A*). This was not the case with the combinations involving W (*Figure 6—figure supplement 1B–C*) where the differentially spliced genes in MW and TW had substantial overlap with the differentially spliced genes in W. However, these synergistically spliced genes were generally distinct from the SEGs (*Figure 6A* and *Figure 6—figure supplement 1D–F*). Despite this distinction, the number of synergistically spliced genes correlated with the EOB score (Pearson r = 0.73, p=0.002; Spearman $r_s$ = 0.75, p=0.0017) over all treatments and time points (*Figure 6B*), as with the SEGs (*Figure 2D*). These data suggest that expression and splicing represent two separate mechanisms driving phenotypic synergy.

## Synergistic activation of transcription factors

We next examined how regulation of the transcriptome can be affected synergistically. We focused on the MCF7 data for this analysis as we were able to leverage a robust pre-existing MCF7-specific transcriptional network (*Woo et al., 2015*). Research has shown that the activity of a transcription factor (TF) can be inferred from expression of its targets (*Lefebvre et al., 2010*; *Alvarez et al., 2016*). Because activity of a TF may be affected in many ways, including post-translational modification, co-factor binding, and cellular localization, this approach is a more robust measure of activity beyond simply measuring expression of the TF itself. We utilized a conservative method for this analysis that distinguishes positive effector and negative effector (repressor) functions of a TF (*Figure 7—figure supplement 1*; see Materials and methods). Of the 1101 TFs studied, most of the differentially active (DA) ones were uniquely active as a positive effector, suggesting that much of the

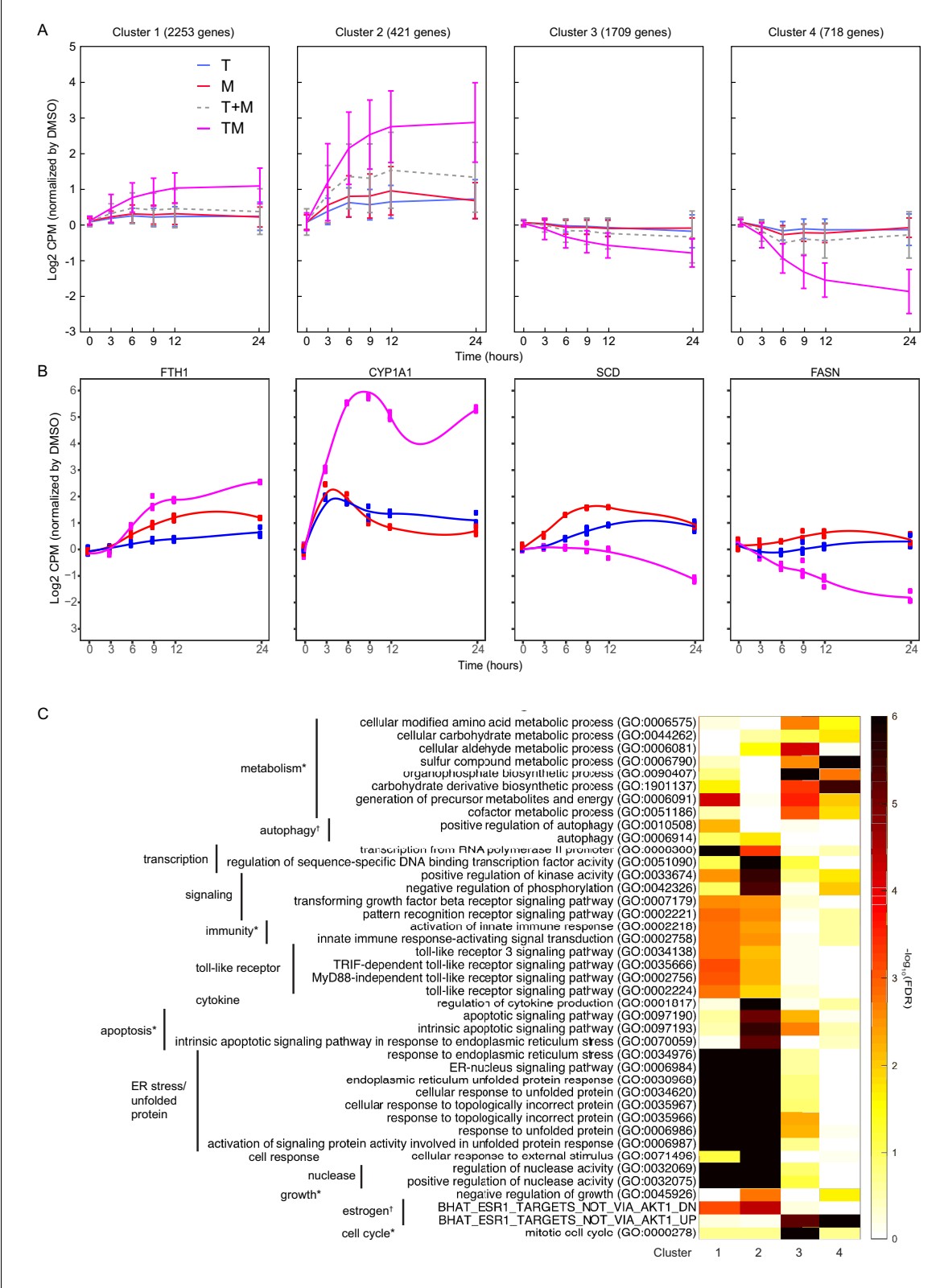

**Figure 4.** Differentially expressed genes have different time courses. (**A**) Mean and standard deviation of gene expression in four clusters identified according to their similarity in expression in T, M, and TM in MCF7 cells. (**B**) Examples of genes in each cluster with significantly different trajectories in TM than the monotherapies. (**C**) Enrichment of the same biological processes as in **Figure 2F** in the clusters. (See also **Source data 7**).

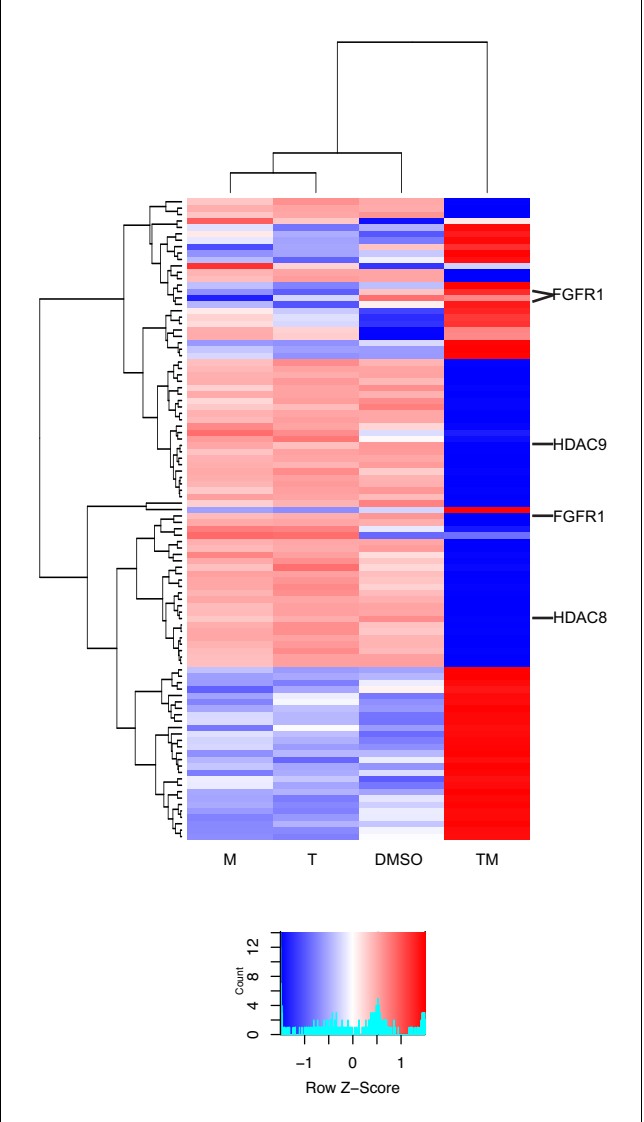

**Figure 5.** New differential splicing emerges in drug combination TM. Top 100 synergistically spliced exons in combination TM at 12 hr in MCF7 cells.

response to these drugs is the result of upregulation of genes, and positive TF-gene interactions (*Figure 7—figure supplement 2A–B*).

Similarly to the differential expression and differential splicing results, most differentially active transcription factors (DATFs) in TM were not DA in the monotherapies T and M (*Figure 7A*). Conversely, most DATFs in TW and MW were also DA in W (*Figure 7B–C*). The majority of DATFs over all treatments and times were produced from genes that were differentially expressed or differentially spliced (*Figure 8A*). However, some instances of DATFs did not correspond to differential expression or splicing and may represent TFs that become DA by mechanisms not captured by RNA-seq, including some that have a known connection to cancer treatment or to biological processes identified in *Figure 3*. For example, ATF4, one of the top DATFs in TM, is not differentially expressed nor spliced, and is a key regulator of the response to endoplasmic reticulum stress (*Pakos-Zebrucka et al., 2016*).

Examining TF activity over time, we found that most DATFs, once DA, tend to remain so at later time points. This time course in TM was distinct from T and M (*Figure 8B*), whereas those of W, TW, and MW were very similar (*Figure 8—figure supplement 1*). In addition, the patterns of differential TF activity were remarkably similar in T and M, and in fact these two monotherapies had a higher

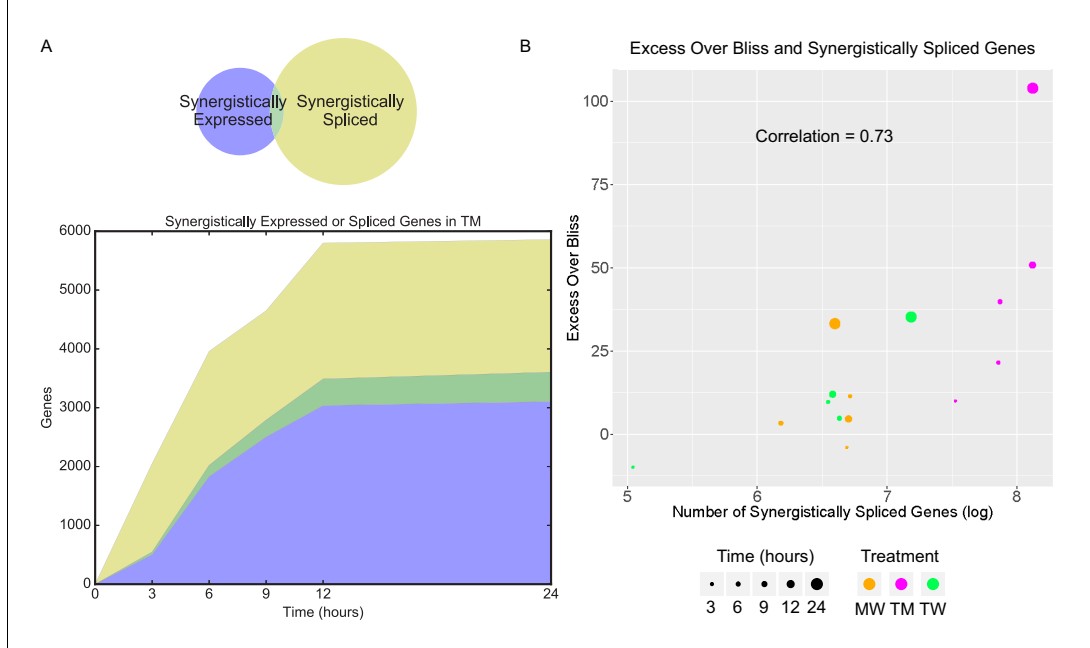

**Figure 6.** Synergistic splicing is distinct from differential expression and associated with synergy. (**A**) Number of synergistically expressed and synergistically spliced genes in TM in MCF7 cells over time; shaded areas correspond to the Venn diagram for 3 hr. (**B**) Relationship of Excess Over Bliss score with the number of synergistically spliced genes in MCF7. (See also *Supplementary file 9*).

The online version of this article includes the following figure supplement(s) for figure 6:

**Figure supplement 1.** Differential and synergistic splicing in MCF7 cells.

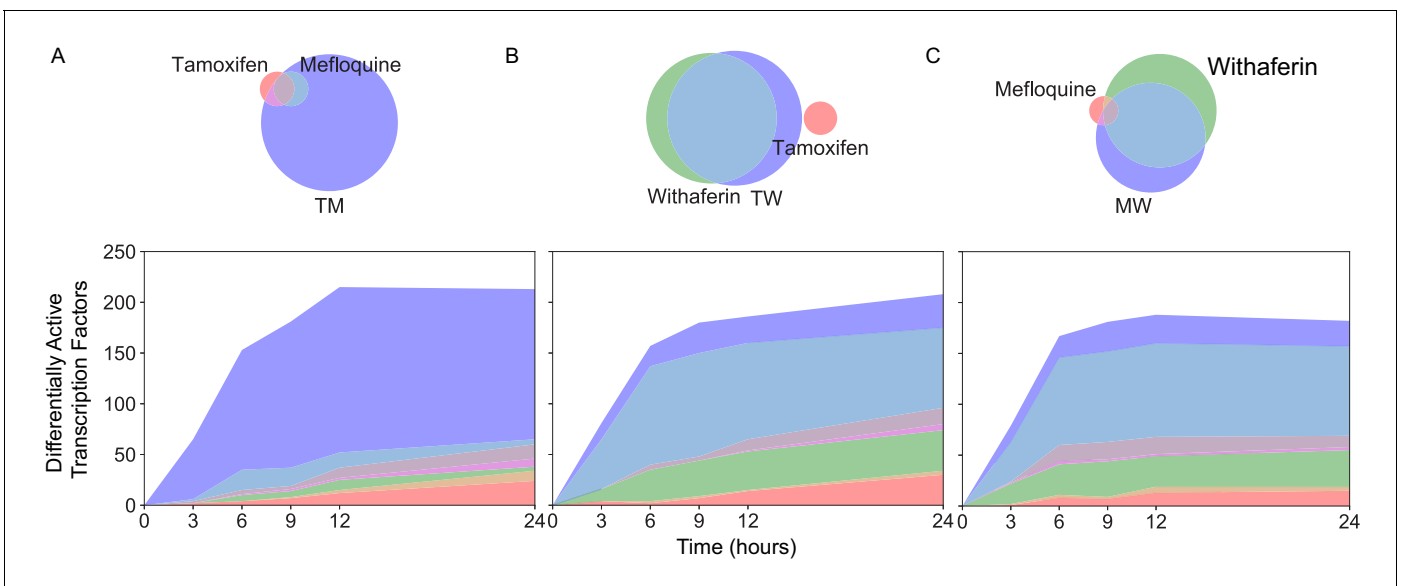

**Figure 7.** New differentially active transcription factors emerge in combination TM. Number of DATFs over time with Venn diagrams at 3 hr in (**A**) T, M, and TM; (**B**) T, W, and TW; and (**C**) M, W, and MW in MCF7 cells. Area represented in each color matches the number of genes in the corresponding color of the Venn diagram; blue areas represent synergistic TFs. (See also *Source data 9*).

The online version of this article includes the following figure supplement(s) for figure 7:

**Figure supplement 1.** Possible changes to transcription factor activity.
**Figure supplement 2.** Classes of differentially active transcription factors in MCF7 cells.

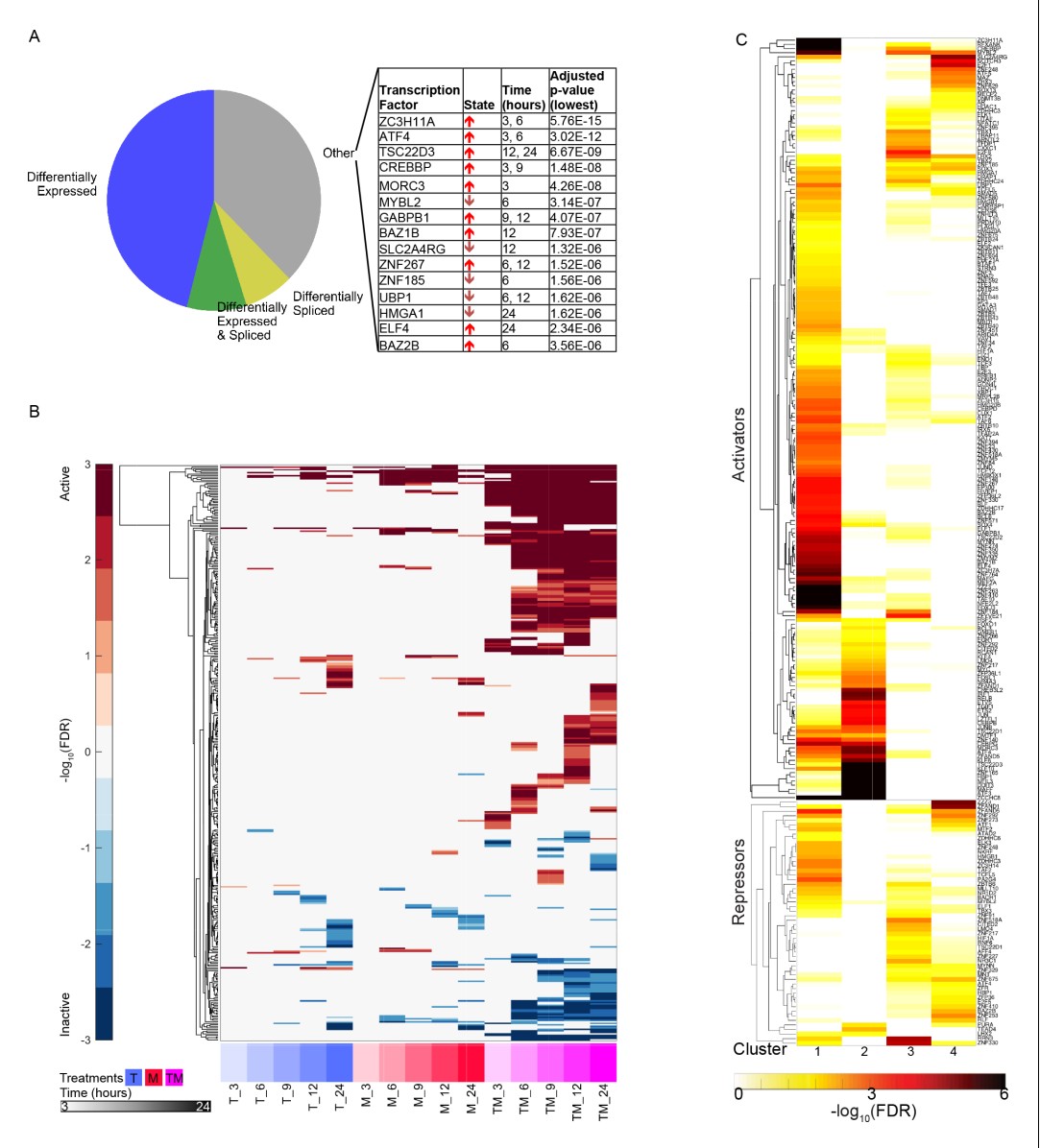

**Figure 8.** Characteristics of differentially active transcription factors. (**A**) All instances of DATFs in MCF7 cells according to the differential expression or splicing status of each TF in the corresponding treatment and time point. The top 20 most significant DATFs not differentially expressed nor spliced are listed. All 20 are positive effectors. Arrows: up = activated, down = inactivated. (**B**) Heatmap of DATFs over time in T, M, and TM at 3–24 hr in MCF7 cells. Color intensity reflects the degree and direction of enrichment by Fisher's Exact test with red for activation and blue for inactivation. Only significant instances are shown. (**C**) Enrichment of gene clusters from *Figure 2F* with sets of TF targets. Color intensity reflects the degree of enrichment by Fisher's Exact test. (See also *Source data 9*).

The online version of this article includes the following figure supplement(s) for figure 8:

**Figure supplement 1.** Differentially active transcription factors in W combinations for MCF7 cells.

correlation in differential activity values of significant TFs than either of the W pairings (Spearman r at 12 hr: 0.8 in T and M, 0.1 in T and W, 0.4 in M and W), echoing the differential expression data (*Figure 2E*). Using the set of DATFs in at least one time point in T, M, and TM (*Figure 8B*), we examined the enrichment of their target sets in the temporal gene clusters identified by k-means clustering (*Figure 4*). The genes in each cluster are significantly enriched in distinct TF target sets, suggesting that the temporal patterns are regulated by different TFs (*Figure 8C*).

## TF activation in monotherapies can account for synergistic gene expression in combinations via a TF activation cascade

We next asked how the combination of T and M gives rise to the synergistic activity of TFs in TM in MCF7. We hypothesized that DATFs in T and/or M could alter the activity of other TFs when both drugs are administered together. We examined two possible mechanisms by which this could happen in combination TM. First, *distinct* DATFs in each monotherapy may converge as regulators of other TFs when the two monotherapies are combined. This is an 'AND' model for the activation of a TF, in that both TFs need to be active in the combination for the activation of their targets. Alternatively, *the same* DATF in T and M may be more strongly DA in TM due to the combined activating effects of the two monotherapies. This dose enhancement mechanism in the combination will be called the 'double-down' model. We also assessed TFs that are linked through the MCF7 transcriptional network to those explained by these AND and double-down models in the same time point, as multiple rounds of transcriptional effects could occur within 3 hr (*Figure 9—figure supplement 1*; *López-Maury et al., 2008*).

We examined the potential effect of these two mechanisms on synergistic TFs and SEGs (*Figure 9* and *Figure 9—figure supplement 1*). At each time point, we identified the synergistic TFs that could have resulted from the AND mechanism (converging red and blue arrows in *Figure 9*), and the double-down mechanism (magenta arrows). At time 3 hr, for example, there are two TFs that are active in T, M, and TM: *MYC* and *KLF10*. These TFs are connected through the network to 9 TFs (*Figure 9*) active in TM (but not in T or M). These TFs are in turn connected to 16 additional TFs (*Figure 9—figure supplement 1*) active in TM (but not in T or M), giving a total of 25 TFs accounting for 42% of all synergistic TFs. Because there was no active TF in T alone, there was no AND mechanism at work. At 9 hr, two new TFs become active in T alone, 18 in M alone and three in both T and M, accounting for 12 new synergistic TFs: six via the AND mechanism (purple), four via the double-down mechanism, and two additional TFs due to a combination of AND and double-down models (*Figure 9—figure supplement 1*). At each time point after 3 hr, we identified TFs connected to TFs identified at the immediately previous time point (vertical arrows). Over all time points, the double-down model alone can explain 83 synergistic TFs, the AND model only explains 12 TFs, and mixed AND and double-down explain 4. In total, this cascade of TF activation accounted for the majority of synergistic TFs at all time points, with 88% of the synergistic TFs at 24 hr explained by the cascade of activation initiated by the 2 TFs activated at 3 hr in both T and M. The number of TFs arising from TFs synergistically activated at previous time points was substantial and accounted for the majority of identified TFs after 3 hr.

We next asked how the AND and double-down mechanisms, along with the activation of synergistic TFs resulting from them, affected the larger group of SEGs. Here, we identified genes potentially affected by the AND and double-down mechanisms, as well as those connected to the newly identified TFs at the current and previous time point. At 3 hr, 29 SEGs can be ascribed to the double-down mechanism, and 146 are direct targets of the newly explained TFs (*Figure 9*). In all, this accounts for 175 (46%) of all SEGs. By 12 and 24 hr, the vast majority (78% and 79% respectively) of SEGs were explained by this cascade. Together, these data suggest that T and M act in concert, mostly through the double-down mechanism, to trigger a transcriptional cascade that results in substantial differential activation of synergistic TFs and genes not seen in either monotherapy.

## Predicting drug synergy in an independent dataset

We have observed that correlation of gene expression of monotherapies is associated with phenotypic synergy in MCF7 cells treated with our three combinations (*Figure 2E*). We next wished to test the generalizability of this association by leveraging the independent DREAM Challenge dataset (*Bansal et al., 2014*), which utilized microarray data from LY3 DLBCL cells treated. Of the 91 drug pairs, 81% of the synergistic combinations have correlation >0.3 (*Figure 10A*). Indeed the average correlation for the pairs with EOB >2.5 (at which the average EOB in three replicates is larger than zero by more than the standard error) is 0.48, which is statistically significantly (t-test p=1.0E-8) larger than the average correlation of 0.3 for pairs with EOB <−2.5. As in our dataset, monotherapy correlation is associated with EOB (Pearson r = 0.27, p=0.009; Spearman $r_s$ = 0.27, p=0.01).

As previous work has suggested that transcriptionally similar, but structurally different drugs are associated with PLD (*Sirci et al., 2017*), we assessed enrichment of the PLD gene signature in the

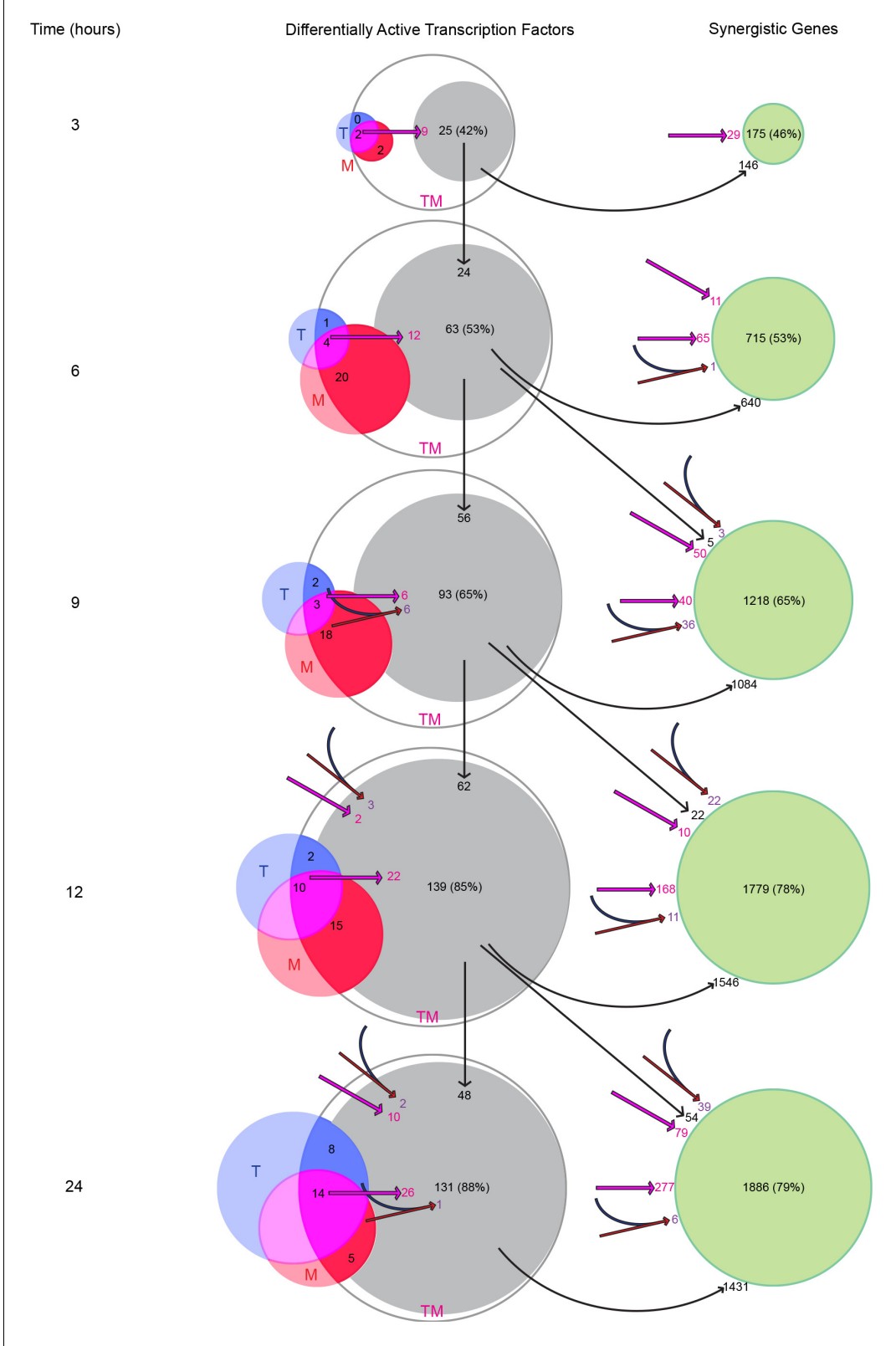

**Figure 9.** Transcription factors become differentially active in a time-dependent cascade in TM. The number of DATFs or SEGs at 3–24 hr are shown as bubbles. Blue, red, and white bubbles represent DATFs in T, M, and TM, respectively. TFs (gray bubbles) and SEGs (green bubbles) shown are 'explained' by the following mechanisms: double-down mechanism at the same (magenta arrow and number) or previous (angled magenta arrow) time point, the AND mechanism at the same (converging blue and red arrows and purple number) or previous (angled converging blue and red arrows) time

*Figure 9 continued on next page*

*Figure 9 continued*

point, or by connection to another TF 'explained' by one of these mechanisms at the same (see supplement 1), or previous (vertical arrows) time point. The total number and percentage of TFs or SEGs in TM meeting any of these criteria is shown. (See also *Source data 10*).

The online version of this article includes the following figure supplement(s) for figure 9:

**Figure supplement 1.** Cascade of differential transcription factor activity.

DREAM drug pairs, using the union of the DEGs in either monotherapy for each pair (see Materials and methods). We found that monotherapy correlation is associated with enrichment in PLD (Pearson r = 0.28, p=0.008; Spearman $r_s$ = 0.27, p=0.009), confirming prior studies. However, we found no association between enrichment in PLD and EOB (Pearson r = −0.10, p=0.36; Spearman $r_s$ = −0.12, p=0.26). Additionally, as in our own dataset, the relationship between correlation and EOB holds when PLD genes are removed (Pearson r = 0.27, p=0.009; Spearman $r_s$ = 0.28,

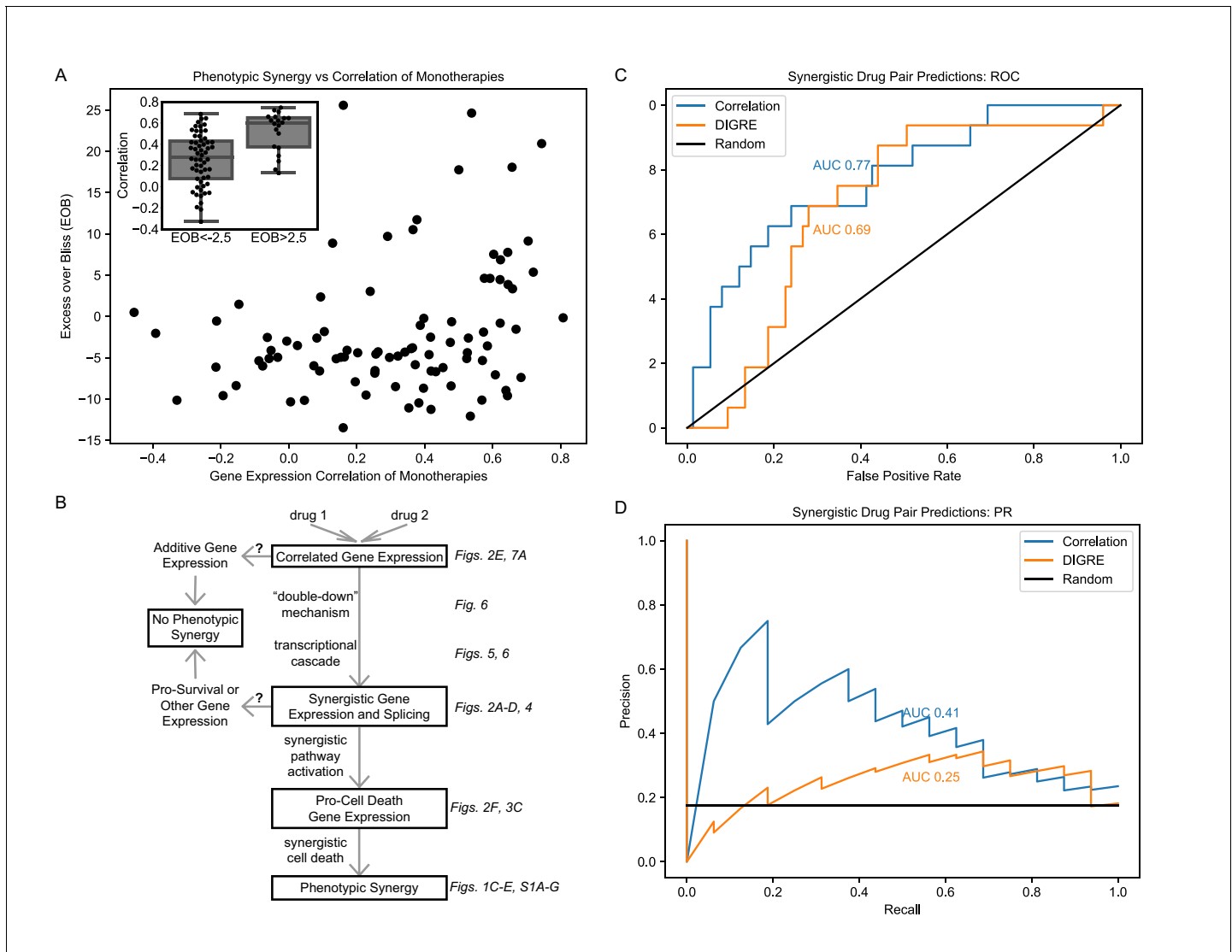

**Figure 10.** Correlation of monotherapies is associated with synergy in an independent dataset. (**A**) Relationship between Excess Over Bliss (EOB) for 91 drug pairs and the correlation between the gene expression of LY3 DLBCL cells treated with corresponding monotherapies in the DREAM dataset. The inset indicates the distribution of correlations for pairs with EOB <−2.5 and EOB >2.5. (**B**) Hypothetical model for the relationships between monotherapy correlation, SEGs, and synergy. Boxed nodes represent phenomena we directly measured in this study. (**C–D**) ROC (**C**) and PR (**D**) for classification of synergistic drug pairs using expression correlation and DIGRE.

p=0.008). These data suggest that correlation of monotherapies may be a necessary, but not sufficient, condition for synergy. Additionally, gene expression in processes such as PLD may be significant, but additive or mildly synergistic in nature (*Figure 3—figure supplement 2*), and thus may play a role in the non-synergistic outcomes of correlated monotherapies. *Figure 10B* outlines a conceptual framework for this relationship. However, we note that unlike in our dataset that is more limited in breadth, correlation of monotherapies and EOB are not linearly correlated in the DREAM data (*Figure 10A*). Furthermore, any relationship between PLD and either molecular or phenotypic synergy has not been explicitly examined in prior studies. Therefore, validation of our findings in multiple contexts is needed to generalize these claims.

Finally, we used the Pearson correlation between DEGs in monotherapies to predict synergy of their combination and compared these results to those of DIGRE, the best performing method in the DREAM Challenge (*Goswami et al., 2015*), in predicting the 16 synergistic drug pairs out of the total 91 pairs in the DREAM dataset (see Materials and methods). Correlation outperforms DIGRE in AUROC (*Figure 10C*) and AUPR (*Figure 10D*), with Bayes factors (*Berger and Pericchi, 2014*) of 3.79 and 34.71, indicating statistical significance with Bayes factors > 3 (*Kass and Raftery, 1995*). It is interesting that simply computing the correlation coefficient between the transcriptomic response of cells to each of a pair of drugs produces a robust predictor of the synergy of the combination.

## Discussion

In this paper, we studied gene expression data taken from cells after treatment with monotherapies and their combinations in a detailed time course analysis, to elucidate the transcriptional mechanisms underlying synergistic drug interactions. We studied three drug combinations on MCF7 breast cancer cells and LNCaP prostate cancer cells: tamoxifen and mefloquine (TM), tamoxifen and withaferin (TW), and mefloquine and withaferin (MW). Of these three combinations, TM was dramatically synergistic (*Figure 1C,F*). A mechanistic rationale for its efficacy is not obvious from the known target processes of T (estrogen signaling; *Shiau et al., 1998*) and M (autophagy; *Sharma et al., 2012*). However, the effect of M on estrogen receptor target gene sets (*Figure 3* for MCF7, *Figure 3—figure supplement 1B* for LNCaP) indicates a moderate anti-estrogen effect, akin to the effect of recently developed novel quinolone derivative estrogen receptor antagonists (*Tang et al., 2016*; *Li et al., 2019b*; *Tang et al., 2014*). This suggests an unexpected overlap in the targets of T and M, even if estrogen receptor represents an 'off-target' of M, rather than a primary target, and may contribute in part to the high gene expression correlation we observed. Although these data, the *in vitro* synergy of TM, and mouse *in vivo* response to chloroquine and tamoxifen (*Cook et al., 2014*) make this combination an attractive candidate for further study, we are not aware of clinical studies on it in cancer. The experiments required to validate the targets and synergistic mechanisms of TM and its *in vivo* effect would be beyond the scope of this study. Our aim was to shed light on the transcriptional response of the combination in terms of the monotherapies.

We have explored the regulation of synergy in MCF7 by integrating our gene expression data with an MCF7-specific transcriptional network, which allowed us to estimate the differential activation of TFs. Possibly due to overlapping target sets of T and M, we find that TF activity is remarkably correlated between T and M treatments at all time points, resulting in a considerably higher correlation in gene expression for this drug pair than the other two drug pairs we examined. This correlated TF activation in response to T and M results in a 'double-down' effect in the combination, where a TF activated by both drugs is reinforced in its activation in the combination at early time points, beginning with early response TFs MYC and KLF10 (*Tullai et al., 2007*). *MYC* is a proto-oncogene present in a low-level amplification in MCF7 cells, likely functioning as an oncogene (*Rummukainen et al., 2001*; *Shimada et al., 2005*; *Shadeo and Lam, 2006*). It has been implicated in regulating the unfolded protein response after prolonged tamoxifen treatment (*Shajahan-Haq et al., 2014*), suggesting it may play a role in the ER stress we observed in response to tamoxifen treatment. Conversely, *KLF10* is a tumor suppressor that represses *MYC* expression in healthy cells and is involved in repressing proliferation and inducing apoptosis (*Ellenrieder, 2008*). It may therefore act to check unregulated *MYC* expression and facilitate the induction of apoptosis.

The action of these TFs triggers a transcriptional cascade that expands over time and results in the emergence of a massive number of SEGs (DEGs in the combination but not in either monotherapy) not recapitulated by increasing monotherapy dose. We found that a high number of SEGs is

strongly associated with the synergistic combination in MCF7, whereas all combinations produced SEGs in LNCaP, only one of which resulted in phenotypic synergy (*Figure 2D*). The data suggest that SEGs are a sensitive, but not specific, molecular indicator of synergistic processes in the combination, which in the case of TM includes pro-cell death processes. This phenomenon is distinct from the relationship between differential expression and cell death, which can reflect processes triggered by single agents, unrelated to the behavior of combinations.

The SEGs resulting from this cascade contribute to specific biological processes that are likely responsible for the killing effect of the combination TM, including activation of intrinsic apoptosis in response to endoplasmic reticulum stress and cell cycle arrest. While T promotes apoptosis (*Chen et al., 1996*), cells are rescued in part by the pro-survival activation of autophagy (*Samaddar et al., 2008*), which degrades and recycles metabolites and other cellular constituents including depolarizing mitochondria (*Klionsky and Emr, 2000*). Autophagy is enriched in T treated cells, likely in response to the unfolded protein response triggered by ER stress (*Klionsky and Emr, 2000*; *Figure 3*, *Figure 3—figure supplement 1B*), and perhaps accounting for the poor efficacy of T in the first 24 hr (*Figure 1C,F*). Research indicates that treatment with M (an antimalarial agent) alters regulation of autophagy in a cell-type specific manner (*Sharma et al., 2012*; *Shin et al., 2015*; *Shin et al., 2012*). This effect may compromise mitochondrial recycling resulting in lower ATP levels (*Wang et al., 2011*).

When treating cells with T and M simultaneously, we expect that the pro-survival effects of the autophagy pathway will be abrogated by M, leading to a synergistic shutdown of metabolic processes, and cell death by apoptosis (*Boya et al., 2005*). Indeed, we observed synergistic changes in apoptosis and autophagy in both cell lines upon TM treatment (*Figure 3*, *Figure 3—figure supplement 1B*). The large numbers of SEGs we observed in all combinations in LNCaP cells (*Figure 2D*) allowed us to study the distinction between SEGs in combinations with phenotypic synergy and those without. W, TW, and MW all quickly downregulated biogenesis of RNA and protein, TW upregulated RNA metabolism, and MW upregulated cholesterol metabolism, which may be physiologic responses to ER stress (*Faust and Kovacs, 2014*; *Christian and Su, 2014*), whereas cells in TM regulated these processes more slowly. Perhaps through the combined protective effects of synergistically upregulating autophagy and downregulating biogenesis, cells treated with the W combinations were able to recover from an initial upregulation of intrinsic apoptosis in response to ER stress. In TM, however, the upregulation of autophagy and downregulation of biogenesis were not as robust, and cells gradually and synergistically downregulated metabolism and upregulated intrinsic apoptosis in response to ER stress (*Figure 3—figure supplement 1A*). These data indicate that drug combinations can induce SEGs that represent either protective responses or cell death, only the latter of which results in phenotypic synergy. This suggests SEGs are a necessary, but not sufficient, condition for synergy.

In MCF7, cell competition induced by varying levels of the synergistically active TF MYC may activate the synergistically upregulated *IRAK2* and its NFkB effectors (*NFKB1*, *CASP8*, and *NFKBIA* are upregulated in TM), triggering apoptosis in the less fit cells (*Meyer et al., 2014*). Cells that are dying or undergoing ER stress produce damage-associated molecular patterns (*Schaefer, 2014*), which activate *TLR3* signaling through a *MYD88*-independent (*MYD88* and its adaptor *IRAK1* are synergistically downregulated in TM) and *TRIF*-mediated pathway leading to the activation pro-inflammatory NFkB signaling (*Pandey et al., 2015*). All these biological processes (*TLR3* signaling, *MYD88*-independent and *TRIF*-dependent regulation of cytokines, NFkB signaling) are synergistically enriched in TM. The crosstalk between cellular stress response and innate immune signaling likely accounts for the enrichment of immunity functional classes (*Muralidharan and Mandrekar, 2013*). The emergence of these synergistic functional gene classes was not recapitulated by increasing monotherapy dose (*Figure 3*). Rather, they are unique to the combination. This suggests that a focus on known targets of monotherapies is insufficient to predict the effects of combinations.

The utilization of RNAseq technology allowed us to examine synergistic effects on RNA splicing, an approach not previously available to studies of synergy that used microarrays. Splicing may alter the proteomic function by adding or deleting a key regulatory protein domain (*Chen et al., 2017*). We found considerable activation of alternatively spliced genes in MCF7 cells in the combination TM but not in either monotherapy. It is of interest that these alternatively spliced genes typically were not differentially expressed themselves. We found that like synergistic gene expression, synergistic splicing was dramatically higher in TM than TW and MW. This likely represents a distinct molecular

mechanism of synergy (*Figure 6*), consistent with previous work on transcription and alternative splicing (*Pan et al., 2004*). Exploring this preliminary evidence further, for example with long-read technology (*Chen et al., 2017*), may reveal a stronger role for isoform switching in drug response than previously thought.

While some previous research has indicated that synergy is context-specific (*Lehár et al., 2009*; *Holbeck et al., 2017*; *Held et al., 2013*), the DREAM Challenge results suggested that similarity of monotherapies is associated with synergy. We observed this phenomenon in the present study in an independent context, measuring similarity by transcriptome correlation, and validated the finding in the DREAM dataset (*Figure 10A*). We also examined the role of the previously identified relationship between PLD and similarity of monotherapy transcriptomes (*Sirci et al., 2017*), validating this relationship but finding that it does not appear to mediate the effect of correlation on synergy.

Our findings have led us to hypothesize about general features of synergy. Our analysis indicates that while all synergistic combinations have correlated monotherapies, the converse is not necessarily true: some drug pairs are correlated in gene expression, but do not generate a synergistic effect. Indeed, when we combine two doses of the same drug (sham combination), whose gene expression profiles are by nature correlated, this does not result in appreciably high EOB (*Figure 2E*). This suggests that the mechanism whereby synergy ensues from transcriptionally correlated but not identical drugs has to include the AND type of activation even if the double-down mechanism is dominant, as was the case in the transcriptional cascade of TM (*Figure 9*). Correlated monotherapies and SEGs appear to be related phenomena that are required for synergy, but insufficient to generate it in all cases. We propose a conceptual framework for the relationship between monotherapy correlation, number of synergistically expressed and spliced genes and the activation of key pathways to account for these findings in *Figure 10B*. In this framework, correlated monotherapies can act through the 'double-down' mechanism generating a transcriptional cascade resulting in expression of many SEGs. We hypothesize that where SEGs are enriched in pro-cell death genes as in our MCF7 data (*Figure 3*), this leads to phenotypic synergy. However, there may exist correlated drug pairs that generate only additive gene expression in biological processes such as PLD, or where SEGs appear but represent pro-survival programs or processes unrelated to cell viability, such as in LNCaP cells treated with TW or MW (*Figure 3—figure supplement 2C*). Under these conditions, correlated monotherapies would not result in a synergistic combination. Further studies on this theory in other contexts are necessary. However to our knowledge, only the data presented here, and the DREAM dataset we used, have matched post-treatment expression and viability data, limiting our ability to validate our findings regarding the gene expression patterns associated with synergy.

We have shown that gene expression correlation can be used to predict synergy with higher accuracy than the best performing algorithm in the DREAM Challenge (*Figure 10C–D*). As the DREAM dataset consists of microarrays in a lymphoma cell line, the importance of correlation appears to be independent of both cell type and gene expression measurement technology. Additionally, the monotonicity of our gene expression time course in the combination (*Figure 4A*) indicates gene expression at later time points can be predicted from earlier ones, and synergy can therefore be predicted from a single time point. These rules of synergy could therefore be used as the basis for *in silico* screening of drug pairs for synergy using existing gene expression datasets. This approach may be an efficient and cost-effective precursor to preclinical studies of drug synergy.

## Materials and methods

**Key resources table**

| Reagent type (species) or resource | Designation | Source or reference | Identifiers | Additional information |
|---|---|---|---|---|
| Cell line (*Homo sapiens*) | MCF7 | American Type Culture Collection | Cat No. HTB-22 | RRID:CVCL_0031 |
| Cell line (*Homo sapiens*) | LNCaP | American Type Culture Collection | Cat No. CRL-1740 | RRID:CVCL_1379 |
| Chemical compound, drug | Withaferin A | Enzo Life Sciences | Cat No. BML-CT104-0010 | |

*Continued on next page*

*Continued*

| Reagent type (species) or resource | Designation | Source or reference | Identifiers | Additional information |
|---|---|---|---|---|
| Chemical compound, drug | Mefloquine hydrochloride | Sigma-Aldrich | Cat No. M2319-100MG | |
| Chemical compound, drug | Tamoxifen citrate | Tocris Bioscience | Cat No. 0999 | |

## Cell culture

MCF7 (ATCC HTB-22) cells were obtained from ATCC. Cells were cultured according to manufacturer's recommendations in ATCC-formulated Eagle's Minimum Essential Medium (Catalog No. 30–2003) with 10% heat-inactivated fetal bovine serum, and 0.01 mg/ml human recombinant insulin. LNCaP cells were purchased from ATCC (Cat No. CRL-1740) and stored in liquid nitrogen until use. Frozen vial was quickly thawed in 37C bath, and then cells were washed from DMSO by spinning in 15 ml vial filled with 10mls of PBS. Cells were re-suspended in RPMI media (ATCC, Cat No. 30–2001) supplemented with 10% Fetal Bovine Serum (ATCC, Cat No. 30–2021) and plated into 75 cm cell culture flask (Corning, Cat No. 430641). Growth media was changed every 3–4 days. After reaching confluence, cells were split at a ratio 1:6. Cultures were tested for mycoplasma periodically using MycoAlert (Lonza, Cat No. LT07-701) per manufacturer's instructions.

To split, media was removed, cells were washed with PBS, and trypsin-EDTA mix was added for 5 min. After detachment, cells were washed with growth media, collected into 50 ml vial, spin down at 1000 RPM, suspended in fresh media and plated into 75 cm flasks. Cells were treated with Withaferin A (Enzo Life Sciences BML-CT104-0010), Mefloquine hydrochloride (Sigma-Aldrich M2319-100MG) or Tamoxifen citrate (Tocris 0999) in 0.3% DMSO for the viability time courses (*Figure 1C–H*, *Supplementary files 4–5*) or 0.4% DMSO for dose-response curves (*Figure 1—figure supplement 1B*, *Supplementary file 6*).

## Viability

The cells were plated at 10,000 cells per well in a clear bottom black 96-well plate (Greiner Cat. No. 655090) and a white 96-well plate (Greiner Cat. No. 655083) then they were placed in an incubator. After 24 hr, the plates were removed from the incubator and treated with drugs using the HP D300 Digital Dispenser. After the targeted drug treatment times, 100 µL of Cell-Titer-Glo (Promega Corp.) was added to the wells in the white 96-well plate and shaken at 500 rpm for 5 min. The plate was then read by the Perkin Elmer Envision 2104 using an enhanced luminescence protocol to count the number of raw luminescent units per well. For the black clear bottom 96-well plates, the plate was spun at 300 g for 5 min and all the cell media was removed. Methanol was then added at 200 µL per well and let sit at room temperature for 15 min. The methanol was removed from the wells and 200 µL of PBS with Hoechst 33342 nucleic acid stain at a final concentration of 1 µG/mL was then added to the wells. The plates were then imaged with the GE Healthcare IN Cell Analyzer 2000 that is equipped with a CCD camera. The IN Cell Analyzer software was used to count the number of cells detected in each well to calculate the viability. Three replicates were used for the combination experiments and two replicates for the dose experiments.

## Calculation of phenotypic synergy

### Excess Over Bliss

Suppose a given drug combination XY inhibits $I_{XY}$ percent of the cells, and the X and Y monotherapies inhibit $I_X$ and $I_Y$ percent of the cells respectively. Note that $V_{XY} = (1 - I_{XY})$ is the viability of the cells, i.e. the percentage of cells that survive after administration of drugs X and Y. Then according to the Bliss model of no interaction between drugs $X$ and $Y$, the percentage of viable cells in the cell culture treated with combination $XY$ is expected to be $V_X V_Y = (1 - I_X)(1 - I_Y)$. In this calculation, any negative values of $I$ that is growth promotion rather than inhibition are converted to 0. This value is used for the 'Bliss Additivity' viability in *Figure 1C-H*. As a result, the EOB independence (*Bliss, 1939*) is given as

$$EOB = 100 * (V_X V_Y - V_{XY}) = 100 * (I_{XY} - (I_X + I_Y - I_X I_Y)),$$

which is the difference between the observed and expected inhibitions. EOB can take any value in the interval $[-100,100]$ and a positive EOB implies synergy, a negative EOB implies antagonism and a value close to zero EOB implies additivity. By propagation of errors, the error of EOB is given as:

$$Error_{EOB} = \sqrt{SEM_X^2(1 + I_Y^2 - - 2I_Y) + SEM_Y^2(1 + I_X^2 - - 2I_X) + SEM_{XY}^2}$$

where $SEM$ represents the standard error of the mean of the inhibition by a given drug.

## Combination index

Although it is simple to calculate, the EOB described above has some limitations as a measure of synergy. For example, it may classify the combination of a drug with itself as synergistic. An alternative method to quantify synergy uses as a null hypothesis the Loewe additivity model and the associated quantity combination index (CI) (*Chou and Talalay, 1984*). The calculation of CI requires fitting a dose response curve to monotherapies. Therefore, one needs the inhibition values for different doses of monotherapies. As a result, we could only calculate CI for 12, 24, and 48 hr for the TM combination and 12 and 24 hr for the TW combination (*Figure 1—figure supplement 1A*) and only for viability measured using CellTiter Glo (*Supplementary files 1–2*).

Mathematically, the combination index CI is computed as

$$CI = D_{x1}/D_1 + D_{x2}/D_2,$$

where $D_1$ and $D_2$ are the required dosage of Drug one and Drug two to reach certain effect (percentage cell death in this case) when both drugs administered independently. On the other hand, $D_{x1}$ and $D_{x2}$ are the dosage required to attain the same percentage of cell death when both drug are given in combination. Accordingly, a CI < 1 suggests synergism, CI = 1 suggests additive and CI > 1 suggests antagonism between the drugs. We used the ComboSyn software (*Chou and Martin, 2005*) to compute CI.

## Processing of the RNA-seq data

The cells were plated at a density of 8000 cells per well in a 96-well plate (Greiner Cat. No. 655083) and placed in an incubator. After 24 hr, the plates were removed from the incubator and treated with drugs using the HP D300 Digital Dispenser. The cells were then collected at the targeted time point by removing the media and pipetting 150 µL of Qiagen Buffer RLT into each well. The plates were then frozen and stored at $-80°C$. For RNA extraction, the Qiagen RNeasy 96 kit (Cat. No. 74181) was used with the Hamilton ML STAR liquid handling machine equipped with a Vacuubrand 96-well plate vacuum manifold. A Sorvall HT six floor centrifuge was used to follow the vacuum/spin version of the RNeasy 96 kit protocol. The samples were treated with DNAse (Rnase-Free Dnase Set Qiagen Cat. No. 79254) during RNA isolation. The RNA samples were then tested for yield and quality with the Bioanalyzer and the Agilent RNA 6000 Pico Kit. The TruSeq Stranded mRNA Library Prep Kit (RS-122–2101/RS-122–2102) was then used to prepare the samples for 30 million reads of single end sequencing (100 bp) with the Illumina HiSeq2500. Three replicates were used for the combination experiments and two replicates for the dose experiments (*Supplementary file 6*).

## Generation of gene level count matrix

We aligned raw reads to hg19 reference genome (UCSC) using the STAR aligner (version 2.4.2a) (*Dobin et al., 2013*). We used the featureCounts (*Liao et al., 2014*) module from subread package (version 1.4.4) to map the aligned reads to genes in the hg19 reference genome, which provided us a gene count matrix with 38 samples and 23228 genes (*Supplementary file 8*). To reduce the noise due to low count genes, we kept genes with at least one count in at least three control (DMSO) samples at any time point. We normalized the resulting count matrix using Trimmed Mean of M-values (TMM) method (*Robinson and Oshlack, 2010*). We produced log base 2 of count per million (cpm) after adjusting plates as covariates (*Source data 1, 2, 3*). We used the voom package (*Law et al., 2014*) to model the mean variance trend in our data (*Figure 1—figure supplement 2A–B*).

## Differential expression analysis

We used the limma (*Ritchie et al., 2015*) pipeline for differential expression analysis to compare treatment with DMSO at respective time points. We corrected the p-values into a false discovery rate (FDR) using BH procedure (*Benjamini and Hochberg, 1995*) for multiple testing (*Source data 5*, *6*, *7*). To determine an appropriate FDR cutoff for differential expression, we examined the data for each treatment at time 0. Time '0' represents a treatment of less than 30 min, during which drug is added and the cells are then immediately prepared for RNA collection. This time delay between treatment and RNA collection is likely long enough to allow transcription of immediate early genes. Immediate early gene expression has been shown to be induced within minutes following an external stimulus (*Tullai et al., 2007*). In the MCF7 combination experiments, the majority of the genes with low p-values at time 0 in our data are known immediate early genes (*Tullai et al., 2007*; *Sas-Chen et al., 2012*). We selected an FDR cutoff of $1.0 \ x10^{-18}$ for differential expression (*Figure 2 — figure supplement 1A*), at which the only DEGs at time 0 over all treatments are well-known immediate early genes (*Table 1*). For the dose experiments, in which there were two replicates instead of three and thus lower p values, we selected $1.0 \ x10^{-5}$ as the lowest FDR cutoff which produced at least as many DEGs as the combination experiments at 24 hr in both T and M (*Figure 2 —figure supplement 1B*). For the LNCaP combination experiments, we selected $1.0x10^{-15}$ as the lowest FDR cutoff for which there were no DEGs at time 0 for any treatments. We did not observe any immediate early genes with low p values at this time point in any treatments in LNCaP.

For the principle components analysis, we used the Python package scikit-learn (*Pedregosa et al., 2011*) to calculate the principle components on the the log fold change of gene expression over the corresponding time point in DMSO, as produced by limma, for each treatment and timepoint.

To calculate monotherapy correlation, for each monotherapy pair, we calculated the Pearson correlation between expression of genes that are differentially expressed in either monotherapy (FDR < 0.1).

## Time course gene expression clustering

To identify the sets of genes that exhibit similar responses to T, M, and TM, we clustered their RNA-seq expression profiles. We considered 5101 genes that are differentially expressed for at least one time point (0, 3, 6, 9, 12, or 24 hr) in at least one of the conditions (T, M, and TM). First, we computed the mean expression profile of each gene from the expression values of its three replicates A, B, and C ($\log_2$(cpm)). We then normalized the mean expression profiles of the 5101 genes by their respective response to DMSO.

Because we wanted to cluster genes that have similar response in T, M and TM, we joined the vector of normalized expression values in T, M, and TM for every gene. Thus, we obtain a vector for each gene that contains 18 values (three drugs * six time points). In order to introduce information about the derivative of the expression profiles, we also joined the delta expression value between each pair of consecutive time point (t3-t0, t6-t3, . . . t24-t12) for the three conditions. Therefore, the vectors to cluster contains each 33 values.

We applied a k-means clustering algorithms to group the expression vectors into k groups. In order to identify a suitable value for k, we computed the total within-cluster sum of squares for values of k running from 1 to 20. We then selected k equal to four clusters as we observed that the gain in information obtained with larger values of k was becoming considerably small. We ran 10,000 times the Hartigan-Wong implementation of the k-means algorithm (*Hartigan and Wong, 1979*) provided by Matlab with a maximum number of iterations set to 1000 before selecting the partitioning of the vectors that achieved the smallest total within-cluster sum of squares (*Source data 7*).

## Gene set enrichment analyses

In each treatment and time point, we separately analyzed the sets of upregulated and downregulated genes for pathways and processes using the built-in Fisher's exact test of the Python package Scipy (*Jones et al., 2001*). We assessed enrichment in all gene sets of the GO_Biological_Process database downloaded from the Enrichr (*Chen et al., 2013*) library repository (http://amp.pharm.mssm.edu/Enrichr/#stats) and, to assess the known effects of tamoxifen, we used the gene set of

estrogen receptor related genes downloaded from the Broad Molecular Signatures Database (*Liberzon et al., 2011*; *Subramanian et al., 2005*; *Bhat-Nakshatri et al., 2009*). We only performed the test where both gene sets contained at least three genes and the overlap contained at least two genes; if either criterion was not met, no p-value was returned, and a p-value of 1 was used for display in the figures. We then calculated the false discovery rate (FDR)-adjusted p-values using the Benjamini-Hochberg method available in the Python package Statsmodels (*Seabold and Perktold, 2010*). To select the gene sets that may explain the synergistic gene expression seen in *Figure 2* and suggest biological processes involved in the synergistic drug response (*Figure 1*), we applied four criteria:

1. Synergistic gene sets were defined as those with an FDR less than 0.00001 in at least one time point in TM and less than 0.01 in all time points in TM, but greater than 0.00001 in all time points in TUM or greater than 0.01 in any time point in TUM. TUM refers to the union of genes from T and M that are either upregulated or downregulated.
2. Additive gene sets were defined as those with an FDR less than 0.00001 in at least one time point in TUM and less than 0.01 in all time points in TUM.
3. GO: Keyword searches for terms associated with each of the ten 2011 hallmarks of cancer (*Hanahan and Weinberg, 2011*) were performed in the Gene Ontology online database (*Ashburner et al., 2000*). For each hallmark of cancer, the highest level ontology relevant to it was selected, followed by 1–2 levels of children of that ontology that were connected by the relation 'is_a', 'regulates','positively_regulates', or 'negatively_regulates'. From the gene sets associated with all these ontologies, those with an FDR less than 0.01 in at least one time point in TM were selected. Five hallmarks remained after applying this filter: metabolism, immunity, cell death (only 'apoptosis' was significant), growth, and proliferation (only 'cell cycle' was significant).
4. To assess known drug targets including estrogen signaling as a target of tamoxifen (*Shiau et al., 1998*) and autophagy as a target of mefloquine (*Sharma et al., 2012*), we included four gene sets related to estrogen signaling from Broad Molecular Signatures Database and any gene sets from the GO Process database containing the words 'autophagy' or 'estrogen'. Similarly to the hallmarks of cancer, any of these sets with an FDR less than 0.01 in at least one time point in TM were selected.

Together, the results of these four approaches comprise the gene sets shown in *Figure 3*, *Figure 3—figure supplement 1*, and *Figure 4C*.

We employed a similar approach to assess enrichment in cellular components, with a particular focus on the lysosome. We assessed enrichment in all gene sets of the GO_Cellular_Component database downloaded from the Enrichr (*Chen et al., 2013*) library repository (http://amp.pharm.mssm.edu/Enrichr/#stats) and, to assess the previously reported lysosomotropic effects of tamoxifen and mefloquine, we used the 250 most upregulated and 250 most downregulated genes in treatment with drugs associated with phospholipidosis, kindly provided by the authors of 'Comparing structural and transcriptional drug networks reveals signatures of drug activity and toxicity in transcriptional responses' (*Sirci et al., 2017*). Based on the findings of the same paper, we also created a gene set made up of the targets of the transcription factors TFEB and TFE3 from our MCF7 network and included it in analysis. The same criteria for statistical testing and false discovery rate procedure as above were used. To select gene sets associated with synergy or additivity, we applied three criteria:

1. Synergistic gene sets were defined as those with an FDR less than 0.00001 in at least one time point in TM and less than 0.01 in all time points in TM, but greater than 0.00001 in all time points in TUM or greater than 0.01 in any time point in TUM. TUM refers to the union of genes from T and M that are either upregulated or downregulated.
2. Additive gene sets were defined as those with an FDR less than 0.00001 in at least one time point in TUM and less than 0.01 in all time points in TUM.
3. Phospholipidosis candidates: We selected the top 20 gene ontology gene sets associated with phospholipidosis in Sirci et al. The gene set 'cytoplasmic vesicle' was too large to be included in the Enrichr library, so the gene sets for 'cytoplasmic vesicle membrane' and 'cytoplasmic vesicle part' were included in its place. From the gene sets associated with these cellular component ontologies as well as the gene sets representing the PLD_up and PLD_down gene signatures and the TFEB_TFE3 transcriptional targets (see above), those with an FDR less than

0.01 in at least one time point in T, M, TUM, or TM were selected. Note that unlike in the hallmarks of cancer analysis, we included any additive gene sets here.

Together, the results of these three approaches comprise the gene sets shown in *Figure 3—figure supplement 2*.

### Generation of exon level count matrix

We mapped the aligned reads to an in-house flattened exon feature file in hg19 reference genome build using featureCounts from subread package (1.4.4). The flattened exon feature file was generated based on gtf (hg19) downloaded from UCSC with overlapping exons from the same gene removed (*Supplementary file 9*).

### Synergistic splicing and exon expression

We used the short read splicing caller diffSplice (*Hu et al., 2013*) in the limma package (version 3.24.3) (*Ritchie et al., 2015*) as framework to detect synergistic spliced genes at each drug combination. We kept exons that have at least one read in at least one sample, and normalized the expressed exon counts using the TMM method (*Robinson and Oshlack, 2010*). For each combination treatment i at a given time point j, we tested synergistic exon expression (SEE) in the generalized linear model of:

$$\mathrm{See_{ij}} = \mathrm{combo_{ij}} + \mathrm{DMSO_j} - \mathrm{singlet1_{ij}} - \mathrm{singlet2_{ij}}$$

We performed two statistical tests to detect synergistic exons expression and synergistic spliced genes. For the former, we performed exon level t-statistic test to detect differences between each exon and other exons from the gene, and defined exons with FDR < 0.05 as synergistically expressed (*Figure 5*). For synergistic splicing, we performed Simes test (*Simes, 1986*) for each gene to test the hypothesis of whether usage of exons from the same gene differed, genes with Simes-adjusted p-values<0.05 are defined as synergistically spliced genes.

### Generation of the MCF7 gene regulatory network

The original MCF7 network has been generated by *Woo et al., 2015* using the network inference method ARACNE2 (*Margolin et al., 2006*) and 448 expression profiles for MCF7 cell line from the connectivity map database (CMAP2; RRID:SCR_015674; *Lamb, 2007*). The original network includes 20,583 probes, 1,109 of which are transcription factors, and 148,125 regulatory interactions. The interactions predicted by ARACNE2 are directed, unless an interaction is found between two TFs, in which case two edges are included in the list (TF1 to TF2 and TF2 to TF1). To obtain a network at the gene level, we applied a one-to-one HG-U133A probe to gene mapping (*Lamb, 2007*; *Li et al., 2011*). The mapping file used has last been updated on July 2015 (v3.1.3). We filtered out edges that don't have both nodes present in the mapping list. Finally, in order to determine positive (activation) and negative (repression) interactions, we calculated the Spearman correlation and the corresponding p-value between each TF-target pair in the network. We then corrected the p-values for multiple hypothesis testing and removed edges with low confidence level (FDR < 0.05), which gave us the final network with 9,760 genes, 1101 TFs, and 48,059 regulatory interactions. For each TF in the network, we defined its positively/negatively regulated targets as the genes to which there exist an outgoing edge in the final network with a positive/negative Spearman correlation coefficient.

### Quantifying transcription factor activity

To calculate differential activity for each TF in our network, we examined its putative targets as determined by our network. We utilized a conservative method for this analysis that distinguishes positive effector and negative effector (repressor) functions of a TF (*Figure 7—figure supplement 1*; *Source data 9*). These functions have been found to be distinct (*Foulkes and Sassone-Corsi, 1992*; *Wray, 2003*; *Partridge et al., 2016*). With this analysis in mind, we performed four comparisons in each treatment and time point:

1. Positively regulated targets of TF and upregulated genes
2. Positively regulated targets of TF and downregulated genes
3. Negatively regulated targets of TF and upregulated genes

4. Negatively regulated targets of TF and downregulated genes

For each of these comparisons, we performed Fisher's exact test as described above for gene set enrichment analysis. This resulted in two p-values for each transcription factor: one for its positive effector function and one for is negative effector function. We then applied the Benjamini/Hochberg FDR adjustment to all the resulting p-values over all time points for the treatment.

We used an FDR cutoff of 0.05 to determine differential activity, and we determined the direction of differential activity of each regulon type (*Figure 7—figure supplement 1*) as follows:

1. Positively regulated targets of TF enriched in upregulated genes → positive effector function activated
2. Positively regulated targets of TF enriched in downregulated genes → positive effector function inactivated
3. Negatively regulated targets of TF enriched in upregulated genes → negative effector function inactivated
4. Negatively regulated targets of TF enriched in downregulated genes → negative effector function activated

We then analyzed each set of two FDR values for the same transcription factor, treatment and time point. If positive and negative effector functions were both activated or both inactivated, the transcription factor was labeled concordant. If one effector function was activated and the other inactivated, the transcription factor was labeled discordant. If only one effector function was differentially active, the transcription factor was labeled unique. For 33 transcription factors in 142 cases across all treatments and time points, the same effector function was found to be both activated and inactivated by the above criteria. These nonsensical results were removed from further analysis, and may be due to transcription factors with an unusually large number of targets.

For each of 1101 transcription factors (*Woo et al., 2015*), we identified positively regulated targets and negatively regulated targets using an MCF7-specific transcriptional regulatory network generated by the ARACNe network inference algorithm (*Margolin et al., 2006*). These two sets represent the distinct targets of each TF's positive effector function and negative effector function. We then assessed the enrichment of each set of targets in the lists of upregulated and downregulated genes in each treatment with respect to DMSO. These enrichment results were used to determine whether transcription factors were activated or inactivated with respect to DMSO (*Figure 7—figure supplement 1*). Most transcription factors were uniquely differentially active in their positive effector or negative effector functions, but not both (*Figure 7—figure supplement 2A–B*).

To account for some transcription factors having very similar sets of targets, we performed Fisher's exact test as above for all possible pairs of transcription factor target sets among those that were significant at each treatment and time point. We then applied the Benjamini/Hochberg FDR adjustment to all the resulting p-values. For each case where the FDR was significant, we then performed Fisher's exact test to assess enrichment of the relevant dysregulated genes in each of three sets: the intersection of the two transcription factor target sets, and each of the target sets individually with the intersection excluded. We then applied the Benjamini/Hochberg FDR adjustment to all the resulting p-values over all cases. Finally, where one effector target set was significantly enriched but the other was not, with their intersection excluded, the significant target set was retained as differentially active, using the original FDR values. All the rest were removed from further analysis.

## Graphical representation of the transcriptional cascade

*Figure 9* shows the evolution of the active transcriptional network after introducing the two drugs T and M. *Figure 9—figure supplement 1* provides a more detailed representation of the mechanisms responsible for the activation of the synergistic TFs. In *Figure 9—figure supplement 1*, each oval represents the set of TFs that are activated under T, M, and/or TM (*Source data 10*): (101) indicates the set of TFs active in T and TM but not in M, (011) indicates the set of TFs active in M and TM but not in T, (111) indicates the set of TF active in T, M, and TM. Finally, (001) are TF active in TM but not in T or M. The number next to each oval indicates the number of TFs in that set. In this way we can keep track of the activation of synergistic TFs (001) in TM in terms of the activation of a pair of parent TFs, one in T (101) and one in M (011), or of one parent TF active also in T and M (111), and doing so at each time point. For example, for the time point at 3 hr (*Figure 9—figure supplement 1*), there 2 TFs active in T, M, and TM that belong to set (111), 2 TFs active only in M and TM that

belong to set (011), and no TF active in T and TM but not in M (101). The middle layer of this representation contains three sets of synergistic TFs that are only active in TM, but not in T or M (001) (grey ovals). Each of these three sets include TFs whose regulators (i.e, their own TFs) are at least one TF in (111) (TF activated through the double-down mechanism: left grey oval in the middle layer), or has two or more parents with one in (101) and the other in (011) (TF activated through the AND mechanism: right grey oval in the middle layer), or has three or more parents from sets (101), (111) and (011) (TFs activated through both double down and AND mechanisms: middle grey oval in the middle layer). At 3 hr only the double-down mechanism can explain nine synergistic TFS, which in turn are parents and can explain the activation of 16 additional synergistic TFs, as indicated in the third layer of (*Figure 9—figure supplement 1*). To the right of the construct we just discussed, there are two more pairs of ovals. The first pair of contains an oval with dashed border, indicating the synergistic TFs that were active at the previous time point (t = 0 for 3 hr) and the arrow points to the grey oval that indicates how many synergistic TF ative at 3 hr can be ascribed to the activation of its regulators in the earlier time point. At 3 hr, both sets are empty. In the rightmost pair of ovals, the bottom one represents the set of TFs whose activation cannot be explained using any of the above mechanisms. Finally, the diagrams for 12 and 24 hr show additional sets of ovals representing TFs that belong to set (101), (011) and (111) at the previous time point, and that are needed to explain some of the synergistic TFs at the current time point, and whose numbers are indicated in italics in the middle layer of the diagram.

## Drug synergy prediction in the DREAM dataset

The DREAM expression matrix was downloaded from Synapse (https://www.synapse.org/#!Synapse:syn2785787). To assess PLD in the DREAM data, we used the aforementioned limma (*Ritchie et al., 2015*) pipeline to calculate differential expression in each treatment compared to DMSO. Then we calculated enrichment of the 500 PLD genes (*Sirci et al., 2017*; see above) in the genes with FDR <0.05 in either monotherapy for each drug pair, using Fisher's exact test as described above. For the correlation-based classifier, for each monotherapy pair, we calculated the Pearson correlation between expression of genes that are differentially expressed in either monotherapy (FDR < 0.1) in the NCI-DREAM data. We ranked the resulting correlations in descending order to calculate a ranking of drug synergy. To test performance by the same measures as the NCI-DREAM challenge, we used the AUROC and AUPR for synergistic drug combination. For the AUC analysis, we used the same criteria as in the dream challenge for the definition of phenotypic synergy resulting in 16 synergistic drug pairs out of the total 91 pairs. To compare our method to DIGRE, we computed the Bayes factor (*Berger and Pericchi, 2014*), a bootstrapped performance distribution between two classifiers. A Bayes factor of 2, for example, means that the first classifier outperformed the second at a 2-to-1 ratio. Two methods that have a Bayes factor <3 may be considered statistically indistinguishable (*Kass and Raftery, 1995*).

## Acknowledgements

We thank Mukesh Bansal, James Bieker, Ross Cagan, Jing He, and Carlos Villacorta for useful discussions. We thank the authors of Sirci et al. for sharing the PLD gene signature. We also acknowledge support to Jennifer Diaz from NIH grants T32 GM007280 and NIH-U54OD020353, and to Andrea Califano from U54 CA209997 (Cancer Systems Biology Consortium), S10 OD012351 and S10 OD021764 (Shared Instrument Grants). Research reported in this paper was supported by the Office of Research Infrastructure of the National Institutes of Health under award numbers S10OD018522 and S10OD026880. The content is solely the responsibility of the authors and does not necessarily represent the official views of the National Institutes of Health.

## Additional information

### Competing interests

Andrea Califano: Dr. Califano is founder, equity holder, consultant, and director of DarwinHealth Inc, a company that has licensed some of the algorithms used in this manuscript from Columbia

University. Columbia University is also an equity holder in DarwinHealth Inc. The other authors declare that no competing interests exist.

## Funding

| Funder | Grant reference number | Author |
|--------|------------------------|--------|
| IBM | IBM Research | Jennifer EL Diaz<br>Mehmet Eren Ahsen<br>Thomas Schaffter<br>Gustavo Stolovitzky |
| National Institutes of Health | NIH T32 GM007280 | Jennifer EL Diaz |
| National Institutes of Health | NIH-U54OD020353 | Jennifer EL Diaz |
| National Cancer Institute | U54 CA209997 (Cancer Systems Biology Consortium) | Andrea Califano |
| National Institutes of Health | S10OD012351 and S10OD021764 (NIH instrumentation grants) | Andrea Califano |
| Icahn School of Medicine at Mount Sinai | | Jennifer EL Diaz<br>Mehmet Eren Ahsen<br>Xintong Chen<br>Bojan Losic<br>Gustavo Stolovitzky |

The funders had no role in study design, data collection and interpretation, or the decision to submit the work for publication.

## Author contributions

Jennifer EL Diaz, Data curation, Software, Formal analysis, Visualization, Methodology, Writing - original draft, Writing - review and editing; Mehmet Eren Ahsen, Data curation, Software, Formal analysis, Visualization, Methodology, Writing - review and editing; Thomas Schaffter, Xintong Chen, Software, Formal analysis, Methodology; Ronald B Realubit, Formal analysis, Investigation, Methodology; Charles Karan, Investigation, Methodology; Andrea Califano, Funding acquisition, Methodology, Writing - review and editing; Bojan Losic, Software, Formal analysis, Supervision, Methodology, Writing - review and editing; Gustavo Stolovitzky, Conceptualization, Formal analysis, Supervision, Funding acquisition, Methodology, Writing - review and editing

## Author ORCIDs

Jennifer EL Diaz (iD) https://orcid.org/0000-0002-3574-0127
Mehmet Eren Ahsen (iD) https://orcid.org/0000-0002-4907-0427
Bojan Losic (iD) https://orcid.org/0000-0003-3643-4612
Gustavo Stolovitzky (iD) https://orcid.org/0000-0002-9618-2819

## Decision letter and Author response

Decision letter https://doi.org/10.7554/eLife.52707.sa1
Author response https://doi.org/10.7554/eLife.52707.sa2

# Additional files

## Supplementary files

- Source data 1. Log counts per million of MCF7 cell combination treatment experiments.
- Source data 2. Log counts per million of MCF7 cell monotherapy dose experiments.
- Source data 3. Log counts per million of LNCaP cell combination treatment experiments.
- Source data 4. Archive of MCF7 combination experiments differential expression data.
- Source data 5. Archive of MCF7 dose experiments differential expression data.
- Source data 6. Archive of LNCaP differential expression data.

- Source data 7. k-means clusters assigned to genes.
- Source data 8. Archive of differential splicing data.
- Source data 9. Archive of differential transcription factor activity data.
- Source data 10. Archive of transcription factors involved in the transcriptional cascade.

• Supplementary file 1. Viability data and calculated EOB for TM dose matrices at 12, 24, and 48 hr in MCF7. Actual values of negative inhibition in monotherapies are included in the heatmap at left. Monotherapy inhibition values used to calculate EOB are shown in the table at right (i.e. Drug1_NPI).

• Supplementary file 2. Viability data and calculated EOB for TW dose matrices at 12, 24, and 48 hr in MCF7. Actual values of negative inhibition in monotherapies are included in the heatmap at left. Monotherapy inhibition values used to calculate EOB are shown in the table at right (i.e. Drug1_NPI).

- Supplementary file 3. Time courses viability data of TM, TW, and MW in MCF7.
- Supplementary file 4. Time courses viability data of TM, TW, and MW in LNCaP.
- Supplementary file 5. Viability data and calculated EOB for TM, TW, and MW at 48 hr in LNCaP.

• Supplementary file 6. Viability data for T and M dose and calculated EOB for sham combinations in MCF7.

- Supplementary file 7. Archive of Raw Fastq IDs.
- Supplementary file 8. Archive of raw expression files.
- Supplementary file 9. Exon counts.
- Transparent reporting form

## Data availability

Raw RNAseq data have been deposited in GEO under accession code GSE149428. Code is available at github.com/jennifereldiaz/drug-synergy (copy archived at https://github.com/elifesciences-publications/drug-synergy).

The following dataset was generated:

| Author(s) | Year | Dataset title | Dataset URL | Database and Identifier |
|---|---|---|---|---|
| Diaz JE, Ahsen ME, Stolovitzky G | 2020 | The transcriptomic response of cells to a drug combination is more than the sum of the responses to the monotherapies | https://www.ncbi.nlm.nih.gov/geo/query/acc.cgi?acc=GSE149428 | NCBI Gene Expression Omnibus, GSE149428 |

The following previously published dataset was used:

| Author(s) | Year | Dataset title | Dataset URL | Database and Identifier |
|---|---|---|---|---|
| Bansal M, Yang J, Karan C, Menden MP, Costello JC, Tang H, Xiao G, Li Y, Allen J, Zhong R, Chen B, Kim M, Wang T, Heiser LM, Realubit R, Mattioli M, Alvarez MJ, Shen Y, NCI-DREAM Community, Gallahan D, Singer D, Saez-Rodriguez J, Xie Y, Stolovitzky G, Califano A | 2014 | sub challenge 2, Drug Synergy Prediction | https://doi.org/10.7303/syn2785787 | Synapse, 10.7303/syn2785787 |

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
