## [Decision Letter]

**Acceptance summary:**

The paper proposes provides further evidence in support of the correlation between transcriptional profiles induced by drugs as predictor of drug synergies. A simple idea that was supported by the NCI-DREAM Drug Synergy Prediction Challenge. The RNAseq time profiles of Tamoxifen and Mefloquine, together with the one of the TM combination, are used to validate the hypothesis in two cell lines (MCF7 and LNCaP).

The paper deals adequately with the scarcity of experimental data to test the model (i.e. additional RNAseq longitudinal datasets), the difficulties with the possible toxic effect of the drug at certain doses and the possible activation of phospholipidosis by the two tested drugs.

The results are, interpreted in terms of the specific activation of the underlying gene regulatory networks as the mechanism underlying drug synergies. Based on these ideas an initial drug synergy prediction algorithm is proposed.

**Decision letter after peer review:**

Thank you for sending your article entitled "The transcriptomic response of cells to a drug combination is more than the sum of the responses to the monotherapies" for peer review at *eLife*. Your article has been evaluated by three peer reviewers, and the evaluation has been overseen by a Reviewing Editor and Kathryn Cheah as the Senior Editor.

The general concept of in-depth analysis of drug combination effect on expression in a few cell lines for a few drug is interesting and offers an alternative to large shallow studies with multiple drugs in many cell lines. The technical developments for the prediction of drug synergy are also very interesting and of potential general value. The key problem that require your input, as detailed below, are with the selection of the drugs and the deduction of the associated interpretation of the mechanism of action.

The reports below point out that tamoxifen and mefloquin, even if widely used, are well known lysosomotropic drugs used in very high concentrations. This represents problems of toxicity and implicates mechanisms that might have more to do with the biology of lysosomes that with any proposed mechanism. As detailed below, it is quite possible that the observed transcriptional response will be more directly related to a general lysosomal stress response than to a specific mechanisms of action of the drugs.

I have to add, that the journal's principles are against asking to include additional drugs, with more precise molecular mechanisms of action and well characterised off target profiles. In the study, since even if it will be an obvious solution will represent an amount of work beyond what is reasonable to ask in a revision.

Therefore, before taking a decision I would like to read your rebuttal to the issues raised above: drugs used in toxic concentrations and mechanisms of action related with unspecific lysosomal toxicity.

At the same time it will be good to hear your response to the rest of the questions below, including software accessibility.

Reviewer #1:

Thank you for the opportunity to review the manuscript from Diaz et al. The authors take on an important and unanswered scientific challenge in trying to predict and understand drug synergies from single drug transcriptomic observations. While I do have some comments to the manuscript I'm overall positive to the undertaking and the manuscript.

My only major concern is that data and source code is not available for peer review. It does say in the first page that this is to be made available but I could not find this.

A few general remarks:

– Drugs with more precisely described mechanisms of action could have been chosen to more intuitively rationalize over the discoveries.

– Code for reproducing analyses should be provided.

– How would the authors explain the contradictory observation that drug with somewhat similar gene expression profiles tend to act synergistically, while drugs with very similar gene expression profiles (e.g. two doses of the same drug) tend to not act synergistically, according to the bliss criterion?

Reviewer #2:

This is a good paper that describes investigation of drug combination effect on expression in a few cell lines for a few drug combinations. Efforts are made to predict drug synergy and efforts are made to show that their ideas generalize to other datasets; both of these efforts strengthen the paper.

I had trouble with the flow of the paper (the structure globally, not the writing locally), and how different components of the study were introduced. Overall I felt that the order of discussion points in the Introduction and early Results sections led to alternating ideas of what the paper contained. I had to read nearly half the paper before sorting it out. For example, you start by saying many general things that imply the study encompasses many drug combinations, then you discuss how you sacrifice breadth for depth and focus on M, T, and W, then you discuss correlations and results on larger datasets. All of this fits together, but a more compact summary/guide to the structure of the paper very early in Introduction could help guide the more casual reader (as even this detail oriented reader got a bit confused).

You introduce the study with much unneeded text about being first. Just focus on findings. Language about timeline linked to order does not last (as new studies emerge), change this language to date-linked and prior-study linked statements. This reviewer is not worried about novelty or being first for such an important topic that could well tolerate comparative studies and replication of results.

A lot rests on Figure 7A, but I do not find the relationship all that convincing.

Although Figure 2E has many data points it is based on your limited set of drugs and essentially one synergistic combination. This seems underpowered.

Is there a mix up in labeling Figure 1C, if not I am missing something, as the predictions don't seem to line up with what I expect given the text.

Reviewer #3:

The manuscript addresses an open and important problem in the field of pharmacology that is how to predict synergistic drug combinations and what are the mechanisms behind the synergy. The authors decided to focus on three drugs: Tamoxifen, Mefloquine and Withaferin and tested dose response of these drugs either alone or in pair-wise combinations and performed in depth transcriptomics studies at different time points. One of the main conclusions of the manuscript is that drugs with transcriptional responses that are correlated tend to be synergistic. Mechanistically the authors try to explain synergy as a caused by transcription factor cascades that are activated only in synergistic drug combinations.

My major concern is that two of the three drugs used, that is Tamoxifen and Mefloquine are well-known lysosomotropic drugs (see references below) that accumulate in lysosomes, especially at the very high concentrations as the ones used in this study, inducing large transcriptional changes. Also, it has been already reported that lysosomotropic drugs, including mefloquine, tend to have correlated expression profiles (Sirci et al., 2017). Also, somewhat troubling is that the only consistently synergistic combination was TM that is two lysosomotropic drugs at large dosages. The authors never mention property such as lysosomotropism and phospholipidosis but they really need to address this point and refer to the relevant literature (see below for some suggestions).

The authors try to address the problem that the synergistic effect they observed may be due to a dosage issue and not to the drug combination, so they treated the 25μM concentration of T in cells as synergistic of 5μM and 20μM and they indeed observed an EOB of 31.5 but not that high as the TM combination with EOB of 99.3. The TM combination, however, is 20μM T and 10μM M (if I understood correctly) so the total dosage is 30μM and it would have been nice to see the effect of T monotherapy to this concentration (and not to 25μM).

Unfortunately, despite the important problem and in depth analyses, the use of these two lysosomotropic drugs at this very large dosages makes the whole work less generic and a less appealing.

Also the overall conclusion that correlated transcriptional responses are a good predictor of synergy, is not that original as the authors themselves stated in the Introduction, similar conclusions were drawn in the DREAM challenge.

Useful references that should be cited in the manuscript:

Nioi et al., 2007; Ellegaard et al., 2016; Nadanaciva et al., 2011; Sirci et al., 2017; Petersen et al., 2013.

---

## [Author Response]

The general concept of in-depth analysis of drug combination effect on expression in a few cell lines for a few drug is interesting and offers an alternative to large shallow studies with multiple drugs in many cell lines. The technical developments for the prediction of drug synergy are also very interesting and of potential general value. The key problem that require your input, as detailed below, are with the selection of the drugs and the deduction of the associated interpretation of the mechanism of action.The reports below point out that tamoxifen and mefloquin, even if widely used, are well known lysosomotropic drugs used in very high concentrations. This represents problems of toxicity and implicates mechanisms that might have more to do with the biology of lysosomes that with any proposed mechanism. As detailed below, it is quite possible that the observed transcriptional response will be more directly related to a general lysosomal stress response than to a specific mechanisms of action of the drugs.

As detailed below and in the revised paper:

1) The doses of the drugs were empirically chosen and in our hands had minimal effects as monotherapies, e.g., viabilities close to 100% for 48 hours after treatment, and few gene expression differences with respect to DMSO. Even if the concentrations appear to be high, drugs can vary from one batch to another, and cell lines can vary with passage and technique. Based on our results, at the doses we used, the *effects* of the drugs as monotherapies are highly unlikely to be supraphysiologic.

2) We analyzed phospholipidosis (PLD, an important response elicited by lysosomotropic drugs) in several ways on both our own data and the DREAM dataset. Interestingly, we observed weak but synergistic effects on lysosome biology at the molecular (transcriptomics) level. On the other hand, we found that PLD was highly enriched but weakly synergistic or additive (at the molecular level) in our data, and was associated with monotherapy correlation in the DREAM dataset but not associated with phenotypic synergy. We conclude that while PLD is associated with monotherapy similarity (confirming Sirci et al. results), it does not explain molecular or phenotypic synergy. Furthermore, even where synergistic effects on lysosome biology or PLD appear, research by Ellegaard et al., 2016, referred to us by reviewer #3 below) would suggest that this may actually be beneficial in cancer combination treatment.

I have to add, that the journal's principles are against asking to include additional drugs, with more precise molecular mechanisms of action and well characterised off target profiles. in the study, since even if it will be an obvious solution will represent an amount of work beyond what is reasonable to ask in a revision.

Unfortunately adding more drugs and their time-resolved transcriptional profiles would not be feasible for this revision. However, we want to highlight that one of the important messages of the paper is the statement that “The transcriptomic response of cells to a drug combination is more than the sum of the responses to the monotherapies” as the title of the manuscript indicates. In that, we believe that the value of our paper adds considerably to the literature.

Having said that, if lysosomotropism were one of the important reasons for similar gene expression (as mentioned above, this is not what we found), then the conclusion would be that what we are observing is an additive effect. But this was addressed in our estimation of the Combination Index (CI), which quantifies synergy factoring out the dose effects. The CI of Mefloquine and Tamoxifen is 0.61 at 24 hours and 0.23 at 48 hours (as shown in Figure 1—figure supplement 1A).

Therefore, before taking a decision I would like to read your rebuttal to the issues raised above: drugs used in toxic concentrations and mechanisms of action related with unspecific lysosomal toxicity.At the same time it will be good to hear your response to the rest of the questions below, including software accessibility.

We apologize for the delay, motivated by the COVID-19 crises and related disruptions. Code used in this paper is now available on GitHub.

Reviewer #1:[…]My only major concern is that data and source code is not available for peer review. It does say in the first page that this is to be made available but I could not find this.

We apologize that we couldn’t provide the code at submission. Code used in this paper is now available at Diaz, 2020github.com/jennifereldiaz/drug-synergy (copy archived at https://github.com/elifesciences-publications/drug-synergy).

Additionally, raw data from the MCF7 and LNCaP experiments have been uploaded to GEO as noted in the paper. Unfortunately, the raw RNAseq data from the dose experiments was lost due to hard drive failure.

A few general remarks:– Drugs with more precisely described mechanisms of action could have been chosen to more intuitively rationalize over the discoveries.

In hindsight the reviewer is right. We started taking a data driven approach. As one of our cell lines was MCF7, an ER+ mammary cancer cell line, we decided to use Tamoxifen because it has been extensively studied as an inhibitor of the estrogen receptor pathway, and because it is clinically the standard of care for ER+ breast cancer. The other drugs were chosen based on the robustness of phenotypic synergy with Tamoxifen, as observed in several replicate experiments. (See subsection “Finding reproducible synergistic and additive combinations” in the Results section.).

– Code for reproducing analyses should be provided.

Code used in this paper is now available on GitHub.

– How would the authors explain the contradictory observation that drug with somewhat similar gene expression profiles tend to act synergistically, while drugs with very similar gene expression profiles (e.g. two doses of the same drug) tend to not act synergistically, according to the bliss criterion?

This is an interesting point. “Self-synergy”, that is the positive excess over Bliss additivity (EOB) that results when two doses of the same drug are administered as in our “sham” combinations (in terminology suggested by reviewer #2), is a synergy that results by perturbing the cell twice in identical directions of transcriptomic response, what we had called in the paper, the double-down mechanism. When a cell culture is treated with two drugs, even when the gene expression is similar, the direction of transcriptomics response is not identical (see response to reviewer 3 for an example of differences in the master regulator MYC, for example), which can give rise to a richer repertoire of cellular response by the AND mechanism.

Reviewer #2:[…]I had trouble with the flow of the paper (the structure globally, not the writing locally), and how different components of the study were introduced. Overall I felt that the order of discussion points in the Introduction and early Results sections led to alternating ideas of what the paper contained. I had to read nearly half the paper before sorting it out. For example, you start by saying many general things that imply the study encompasses many drug combinations, then you discuss how you sacrifice breadth for depth and focus on M, T, and W, then you discuss correlations and results on larger datasets. All of this fits together, but a more compact summary/guide to the structure of the paper very early in Introduction could help guide the more casual reader (as even this detail oriented reader got a bit confused).

Thanks for your suggestion. We have tried to orient the reader with a more compact and earlier guide of the structure of the paper in the last paragraph of the Introduction. Also Figure 7B serves as a summary of the conclusions of the paper.

You introduce the study with much unneeded text about being first. Just focus on findings. Language about timeline linked to order does not last (as new studies emerge), change this language to date-linked and prior-study linked statements. This reviewer is not worried about novelty or being first for such an important topic that could well tolerate comparative studies and replication of results.

We thank the reviewer for this comment. We removed the references to priority and novelty in the Abstract, Introduction, and throughout the paper.

A lot rests on Figure 7A, but I do not find the relationship all that convincing.

There are no other datasets that have a relatively large number of combinations with phenotypic synergy of the combinations and post-treatment gene expression of the monotherapies (that we know of). However, we added text to point out that more examples are needed to claim that these results are fully generalizable (subsection “Predicting drug synergy in an independent dataset” and Discussion, ).

Although Figure 2E has many data points it is based on your limited set of drugs and essentially one synergistic combination. This seems underpowered.

We added a point to say that this is a limited dataset and more drugs are needed to validate the result (subsection “Gene expression of drug combinations in relation to monotherapies”).

Is there a mix up in labeling Figure 1C, if not I am missing something, as the predictions don't seem to line up with what I expect given the text.

We changed “Predicted” in Figure 1C to “Bliss Additivity” and clarified the meaning of this in the figure legend.

Reviewer #3:The manuscript addresses an open and important problem in the field of pharmacology that is how to predict synergistic drug combinations and what are the mechanisms behind the synergy.

We appreciate the reviewer’s comment regarding the open questions addressed by this study. It is also important to note that we also aim to address another question that has not been addressed in the literature, and that is not related to synergy, namely how the transcriptomic response of cells to a drug combination is related to the transcriptional responses to the monotherapies.

The authors decided to focus on three drugs: Tamoxifen, Mefloquine and Withaferin and tested dose response of these drugs either alone or in pair-wise combinations and performed in depth transcriptomics studies at different time points. One of the main conclusions of the manuscript is that drugs with transcriptional responses that are correlated tend to be synergistic. Mechanistically the authors try to explain synergy as a caused by transcription factor cascades that are activated only in synergistic drug combinations.My major concern is that two of the three drugs used, that is Tamoxifen and Mefloquine are well-known lysosomotropic drugs (see references below) that accumulate in lysosomes, especially at the very high concentrations as the ones used in this study, inducing large transcriptional changes.

The reviewer is correct that μM drug concentrations in cell culture are often considered supraphysiologic and high. However, in our study both Tamoxifen (T) and Mefloquine (M) were used at empirically chosen concentrations that didn't affect survival when used as monotherapies. Indeed, Figure 1C shows that the viability of MCF7 cells when treated with T and M independently is close to 100% at all time points up to 48 hours. Figure 1F shows the same is true for LNCaP cells up to 24 hours. Furthermore, there were less than 100 genes differentially expressed by either T or M at 24 hours in MCF7 (Figure 2A) indicating that the transcriptional response to the monotherapies at the doses considered here was at best mild. Another way to see this is in Figure 1I, which shows the heat map of the transcriptional response to DMSO, T and M. It can be seen that at the doses used, the transcriptional profile of T and M is more similar to DMSO than to TM.

The apparent discordance of the fact that other labs may have found that 20 μM of T or 10 μM of M elicits a strong effect can be accounted for, at least in part, by the fact that different viability assays can be difficult to compare even if using the “same” drugs and the same “cell types” as used in other laboratories. The paper “Genetic and transcriptional evolution alters cancer cell line drug response” by Ben-David, et al. in Todd Golub’s lab (cited in the subsection “Finding reproducible synergistic and additive combinations”) showed a wide variety of genetic profiles and gene expression among 27 different versions of MCF7 cells. In addition, their drug experiments seem to indicate that just 7 passages are enough to cause noticeable differences in drug response, including abrogating some drug responses and inducing responses where there were none. The authors argue that this “cannot be explained by irreproducibility of the assay.”

Also it has been already reported that lysosomotropic drugs, including mefloquine, tend to have correlated expression profiles (Sirci et al., 2017).

We thank the reviewer for bringing the interesting Sirci et al.’s paper by Diego di Bernardo’s lab to our attention. There are a number of discoveries in Sirci et al. . that are relevant to our paper. The authors discover that amongst the most “transcriptionally close” and “structurally distant” drug pairs there was an enrichment of lysosomotropic drugs and drugs inducing PLD. They then studied 36 cationic amphiphilic drugs (CAD) drugs that are known to induce PLD, and did a Drug Set Enrichment Analysis (DSEA) analysis, to find that in aggregate these 36 drugs have

“lysosome” as the top enriched GO cellular component. They then compiled the PLD consensus transcriptional signature and found that 50% of the compounds induced PLD at high concentrations. All this suggests that lysosomotropic drugs induce a transcriptional response associated with PLD.

Also somewhat troubling is that the only consistently synergistic combination was TM that is two lysosomotropic drugs at large dosages. The authors never mention property such as lysosomotropism and phospholipidosis but they really need to address this point and refer to the relevant literature (see below for some suggestions).

We appreciate the reviewer pointing us to literature on PLD. This represents an interesting mechanistic insight into drug toxicity. We reviewed Sirci et al.’s paper in detail. Sirci et al. showed that two lysosomotropic drugs that induce PLD elicit a cellular response characterized by a shared transcriptomics signature of about 500 genes. Because of its relevance to our study, we studied the PLD signature effect as well as other possible transcriptional processes enriched in our experiments, and studied the degree to which the PLD signature contributes to the similarity between the transcriptional responses of the cell to the two drugs.

In this revision, we introduced a new set of analyses on phospholipidosis enrichment in our data. We assessed several gene sets based on Sirci, et al.’s data, including the PLD gene signature they derived, kindly provided by Diego di Bernardo upon our request, the top 20 Gene Ontology gene sets they found associated with PLD (which all happen to be cellular components, likely because they did not assess biological processes separately), and the transcriptional targets of two master regulators of PLD they identify, based on our MCF7 network. Our results are described throughout the paper (subsection “Critical cancer pathways are synergistically enriched” and Discussion), and the important new paragraphs are colored in blue in the revision. We further assessed enrichment in the PLD signature in all the drug pairs in the datasets used in the DREAM Challenge and compared it to correlation and synergy (491-507). Taken together, we find:

1) Validation of Sirci, et al.’s findings in independent contexts, in that PLD is highly enriched in tamoxifen and mefloquine treated cells in our data, and monotherapy correlation is associated with PLD enrichment in the DREAM data.

2) PLD appears to be an insufficient mechanism to explain correlated monotherapies. In both our data and the DREAM data, monotherapy correlation was still associated with EOB when we removed the 500 PLD genes from the calculation. In addition, while the

TFEB/TFE3 target set was weakly but synergistically enriched in TM (Figure 3—figure supplement 2B), only TFE3 is activated in the cascade of TFs we analyzed that trigger SEG expression in MCF7, and it only appears at 24 hours (Figure 3—figure supplement 2). Finally, a number of diverse biological processes are synergistically enriched in TM (Figure 2F) that are unlikely to all be explained mechanistically by PLD.

3) We previously noted that correlated monotherapies must in some cases result in additive gene expression, or synergistic gene expression that does not lead to phenotypic synergy (Figure 7B). In the DREAM data, we found that enrichment in PLD is not associated with EOB. We therefore propose that some of the correlated but additive drug pairs may exhibit PLD. In other words, the data suggest that correlated monotherapies are associated with both PLD and synergy, but apparently by different mechanisms, as PLD and synergy are not associated with each other.

The authors try to address the problem that the synergistic effect they observed may be due to a dosage issue and not to the drug combination, so they treated the 25μM concentration of T in cells as synergistic of 5μM and 20μM and they indeed observed an EOB of 31.5 but not that high as the TM combination with EOB of 99.3. The TM combination, however, is 20 μM T and 10μM M (if I understood correctly) so the total dosage is 30μM and it would have been nice to see the effect of T monotherapy to this concentration (and not to 25μM).

The viability at 30μM was measured to be 13.7% but not reported in the previous version of the manuscript because at those low viabilities we couldn’t extract sufficient amounts of good quality RNA from the cell culture for the RNAseq experiments. However, as requested by the reviewer, we are happy to report it in this revision. As shown in Figure 1—figure supplement 1B, the viability after treatment with T at 30 μM was 13.7%. The Excess over Bliss corresponding to the sham combinations at 30 μM (assuming a combination of 10μM + 20μM) was 53.1%, considerably smaller than the EOB of ~100% for TM at comparable doses, and comparable final Viability (subsection “Finding reproducible synergistic and additive combinations”). For the discussion of the transcriptomics of sham combinations we used the RNA extracted from the cell culture treated with T at 25 μM.

Another indication that the reduction of viability in TM is due to synergy and not to dose effect is the Combination Index (CI), which quantifies synergy factoring out the dose effects. CI for Mefloquine and Tamoxifen is 0.61 at 24 hours and 0.23 at 48 hours (as shown in Figure 1—figure supplement 1A). A CI < 1 indicates synergy.

Unfortunately, despite the important problem and in depth analyses, the use of these two lysosomotropic drugs at this very large dosages makes the whole work less generic and a less appealing.

We believe that the above discussion justifies that the dosages are not large enough to obfuscate the MOA of the individual drugs. Our new analyses show that the PLD response is not the only process at play in the response of the cells. We substantially rewrote some parts of the paper to reflect the presence of a PLD enrichment in the transcriptional response to drugs like those we studied, as suggested in Sirci et al.  (subsection “Critical cancer pathways are synergistically enriched”).

Also the overall conclusion that correlated transcriptional responses are a good predictor of synergy, is not that original as the authors themselves stated in the Introduction, similar conclusions were drawn in the DREAM challenge.

The reviewer is correct. In the DREAM Challenge there were many hypotheses as to what is the transcriptomic response of a combination given the individual responses. Some of the best performing algorithms hypothesized that similar monotherapy transcriptomes would result in synergy. In this paper we can answer this question at least for a small subset of drugs. We directly demonstrate that correlated monotherapy transcriptomes are associated with synergy, and suggest that the “double down” mechanism seems to drive this effect. We use DREAM as a secondary validation of the simplest version of this association. We have clarified these relationships between our work and the DREAM challenge (Discussion).

Useful references that should be cited in the manuscript:Nioi et al., 2007; Ellegaard et al., 2016; Nadanaciva et al., 2011; Sirci et al., 2017; Petersen et al., 2013.

We thank the reviewer for pointing out this literature and have added the citations.